



# Methane ebullition as the dominant pathway for carbon sea-air exchange in coastal, shallow water habitats of the Baltic Sea

Thea Bisander[1], John Prytherch[2], Volker Brüchert[3, 4]

[1] School of Natural Sciences, Technology and Environmental Studies, Södertörn University, Huddinge, 141 89, Sweden
[2] Department of Earth Sciences, Uppsala University, Uppsala, 75236, Sweden
[3] Department of Geological Sciences, Stockholm University, Stockholm, 10691, Sweden
[4] Bolin Centre for Climate Research, Stockholm, 10691, Sweden

*Correspondence to*: Thea Bisander (thea.bisander@sh.se)

**Abstract.** Shallow coastal marine habitats are hotspots for carbon dioxide ($CO_2$) and methane ($CH_4$) exchange with the atmosphere, yet these fluxes remain poorly quantified, limiting their integration into global and regional carbon budgets. With the use of floating chambers, this study quantified seasonal and annual $CO_2$ and $CH_4$ fluxes in common Baltic Sea habitats using, including macroalgae-covered coarse sediments, sparsely to densely vegetated sands, submerged plant-covered mixed substrates, and reed-dominated muds. Monthly average $CO_2$ fluxes ranged from $-937 \pm 161$ to $3\ 512 \pm 704$ mg m$^{-2}$ d$^{-1}$, with macroalgae and reed habitats exhibiting distinct flux ranges setting them apart from the sand and mixed substrate habitats. Apart from the macroalgae, all habitats exhibited a net efflux of $CO_2$ on an annual basis. Diffusive $CH_4$ fluxes varied seasonally, from $0.1 \pm 0.01$ to $26 \pm 1.5$ mg m$^{-2}$ d$^{-1}$ with peak emissions in summer. Ebullition fluxes occurred between March and October, reaching up to 232 mg m$^{-2}$ d$^{-1}$ and significantly contributed to, or even dominated, the annual exchange of both $CO_2$ and $CH_4$ in the sand, mixed substrate, and reed habitats. Upscaling these fluxes to the shallow-water ($< 6$ m) zone of the Stockholm archipelago yielded total $CO_2$-equivalent fluxes of between $-0.01$ and $0.2$ Tg $CO_2$-eq yr$^{-1}$ on a 100-year timescale. In comparison, Stockholm's energy- and transport sector emits approximately 1.2 Tg $CO_2$-eq yr$^{-1}$, suggesting that the shallow-water coastal zone could be a small, but significant contributor to the total source strength of the Stockholm region.

## 1 Introduction

Coastal marine environments play a vital role in the global carbon cycle, functioning as atmospheric sources or sinks of carbon dioxide ($CO_2$), and as sources of methane ($CH_4$) (Friedlingstein et al., 2022). The sea-to-air exchange of these gases is governed by the water-air boundary layer conditions, which determine the gas transfer velocity (Liss & Slater, 1974) and the gas saturation of surface waters (Gustafsson et al., 2015). Coastal zones exhibit intense biogeochemical cycling fuelled by high rates of primary production, which, in combination with both terrestrial and riverine inputs of dissolved carbon species (Bauer et al., 2013), facilitates the exchange of $CO_2$ and $CH_4$ with the atmosphere (Resplandy et al., 2024).



30   Vascular plants and algae fix $CO_2$ via photosynthesis as organic material, while respiration and decomposition recycle it back to $CO_2$ (Gattuso et al., 1998). Dissolved $CO_2$ levels are also regulated by alkalinity, which affects the balance of carbonate species and is influenced by riverine input, groundwater input, sediment-seawater, and shelf-coastal exchange (Middelburg et al., 2020). In contrast, $CH_4$ in surface waters is mainly produced via methanogenesis in anaerobic, organic-rich, and sulphate-depleted sediments, and escapes via diffusion or ebullition (Reeburgh, 2007). Around 70 % of coastal $CH_4$

35 emissions are believed to come from sediment ebullition (Weber et al., 2019) which occurs when the partial pressure of the gases in the sediment exceeds their solubility at hydrostatic pressure, leading to release of excess free gas in the form of bubbles. Due to their rapid rise rate, a large fraction of free gas avoids microbial oxidation in the sediment and water column and is transported towards the water-air boundary layer as ebullition flux (Hermans et al., 2024), adding to the diffusive exchange across the sea-air boundary (Mao et al., 2022). While ebullition may be the dominant flux pathway for $CH_4$ in coastal

40 waters, the size of the flux remains uncertain due to its highly stochastic nature (Lohrberg et al., 2020).

   There is increasing interest in using the carbon sequestration potential of the coastal ocean as a climate mitigation tool to help contain global warming as close to the 2015 Paris Agreement as possible (Claes et al., 2022). Emission reduction strategies could potentially benefit from coastal mitigation measures but require accurate quantification of emission or uptake of greenhouse gases in coastal areas. Significant uncertainties remain in the $CO_2$ and $CH_4$ budgets of coastal regions as a

45 consequence of the scarcity of in-situ flux measurements and the difficulties of upscaling existing flux measurements due to the high spatial and temporal variability, both within and between individual habitats (Dai et al., 2022; Bange et al., 2024). A critical step in upscaling $CO_2$ and $CH_4$ fluxes from the coastal zone is a suitable habitat classification system. Simple classifications based solely on single properties such as sediment or vegetation type can fail to capture the full spectrum of habitat diversity and variability leading to inaccurate representations of flux dynamics, while overly detailed classifications

50 risk becoming impractical for data collection and analysis. Moreover, their effectiveness is constrained by the lack of detailed, high-resolution coastal habitat maps (Rosentreter et al., 2023).

   Significant efforts have been made to constrain gas exchange in highly productive environments like mangroves, seagrass meadows, and saltmarshes due to their high potential for sediment carbon sequestration (Rosentreter et al., 2023). However, recent studies in the Baltic Sea indicate that northern, temperate habitats with macroalgae, mixed vascular plant

55 vegetation, and even relatively unvegetated sediments, also contribute significantly to $CO_2$ and $CH_4$ exchange with the atmosphere (e.g. Lundevall-Zara et al., 2021; Asmala & Scheinin, 2023; Roth et al., 2023). Despite this, only a few studies have reported on both $CO_2$ and $CH_4$ fluxes in these habitats (Asmala & Scheinin, 2023; Roth et al., 2023), which is crucial to assess the effect the sea-air exchange has on radiative forcing in the atmosphere. $CH_4$ has a sustained-flux global warming potential that is 45 or 96 times as efficient as $CO_2$ on a 100- or 20-year timescale, respectively, and as such, even relatively

60 small quantities of $CH_4$ can significantly contribute to the exchange (Neubauer & Megonigal, 2015).

   Various methods exist for measuring gas exchange between the water and atmosphere. These are either based on water and air concentration sampling and determination of fluxes through a gas transfer velocity parametrisation (e.g. Humborg et al., 2019; Asmala & Scheinin, 2023; Roth et al., 2022, 2023), eddy covariance (e.g. Gutiérrez-Loza et al., 2019), or floating



chamber techniques (e.g. Lundevall-Zara et al., 2021), each with their own strengths and limitations (Bastviken et al., 2022).
The floating chamber technique has the benefit that the flux is determined directly, avoiding the use of a gas transfer velocity parametrisation, and is capable of resolving the ebullition flux from the diffusive flux component. In addition, since the measurement has a high (analyser-dependent) sensitivity and small footprint, small-scale habitat differences can be resolved, and low fluxes can be included in habitat budgets.

In this study, we conducted year-round floating chamber experiments in shallow (<4 m) coastal habitats in the
Stockholm and Trosa archipelagos of the northwestern Baltic Proper to quantify diffusive $CO_2$ and diffusive and ebullition $CH_4$ fluxes. The objective was to constrain the flux variability based on five habitat groups, including macroalgae-covered coarse sediments, sparsely to densely vegetated sands, submerged plant-covered mixed substrates, and reed-dominated muds. These habitats are commonly occurring along both the Swedish and Finnish Baltic Sea coast (Al-Hamdani & Reker, 2007). Further, the study aimed to quantify the relative contributions from $CO_2$ flux, diffusive $CH_4$ flux and $CH_4$ ebullition to the total
$CO_2$-equivalent flux, identifying the dominant pathway for carbon-based greenhouse gas exchange in these habitats.

## 2 Methods and materials

### 2.1 Sampling locations and habitat classification

The sampling was carried out in seven locations in the Stockholm and Trosa archipelago seas in the north-western Baltic Proper (Fig. 1). Six out of seven sampling locations were on the island of Ingarö and the seventh sampling location was on
the island of Askö.

The archipelagos are brackish systems where the salinity spans from close to zero in the inner archipelago near the outflow from lake Mälaren and up to 8 ‰ in the outer archipelago (Fig. 1). The coastal biotopes consist of exposed rocky coastline, long and narrow fjord-like bays, and sheltered inlets (Hill & Wallström, 2008; Kautsky, 2008). The Stockholm archipelago is considered one of the most eutrophic archipelagos along the Swedish coastline (Hill & Wallström, 2008).
The classification scheme applied in this study is HELCOM HUB, an underwater biotope and habitat classification system developed for the Baltic Sea by HELCOM (HELCOM, 2013). The sampling locations were categorised based on bottom substrate and vegetation (Table 1). The seven sampling locations were divided into five distinct habitats: Coarse sediment with perennial algae cover (location A), sand with sparse epibenthic macrocommunity (location B), sand with submerged rooted plants (location C), mixed substrate with submerged rooted plants (locations D and E) and muddy sediment
with emergent plants Phragmites australis (locations F and G). The classification was completed in September 2020. For further details on the HELCOM HUB classification and how it was carried out in the study, see Supplementary Material Text S1.



**Table 1. Descriptions of the sampling locations A-G as shown in Fig. 1. Max depth refers to the maximum depth that a flux measurement was taken at.**

| Location | Bottom substrate | Vegetation cover | Taxa description | Max depth | Salinity |
|---|---|---|---|---|---|
| A | Coarse sediment | ≥10 % perennial algae | Fucus vesiculosus, Cladophora glomerata. | 1 m | 4.9 – 5.7 |
| B | Sand | 0< >10 % epibenthic macrocommunity | Stuckenia pectinate, Ruppia sp., phytoplanktonic algae blooms in summer. | 4 m | 3 – 5.7 |
| C | | ≥10 % submerged rooted plants | Phragmites australis along shoreline and Stuckenia pectinate, Ruppia spp., Myriophyllum spicatum. | 3 m | 5 – 5.6 |
| D | Mixed substrate | ≥10 % submerged rooted plants | Phragmites australis along shoreline and Stuckenia pectinate, Ruppia spp., Myriophyllum spicatum, Ceratophyllum demersum together with an algal community of Fucus spp. and various filamentous algae | 4 m | 5.9 – 6.3 |
| E | | | | 3 m | 2.3 – 5.2 |
| F | Muddy sediment | ≥10 % Emergent vegetation | Phragmites australis | 1 m | 5 – 5.5 |
| G | | | | 2 m | 2.7 - 5 |

**2.2 Field methods**

Sampling was carried out between September 2020, and October 2022, with seasonal sampling periods broadly in January (Jan 25-26th 2021), March (Feb 27th-Apr 2nd 2021), May (Apr 28th-May 20th 2021), June (Jun 16th-17th 2021), July (Jul 8th-9th 2021), September (Sep 14th-15th 2020, Sep 14th-22nd 2022), October (Oct 26th-29th 2022), and the shift between November and December (Nov 25th-Dec 10th 2020).






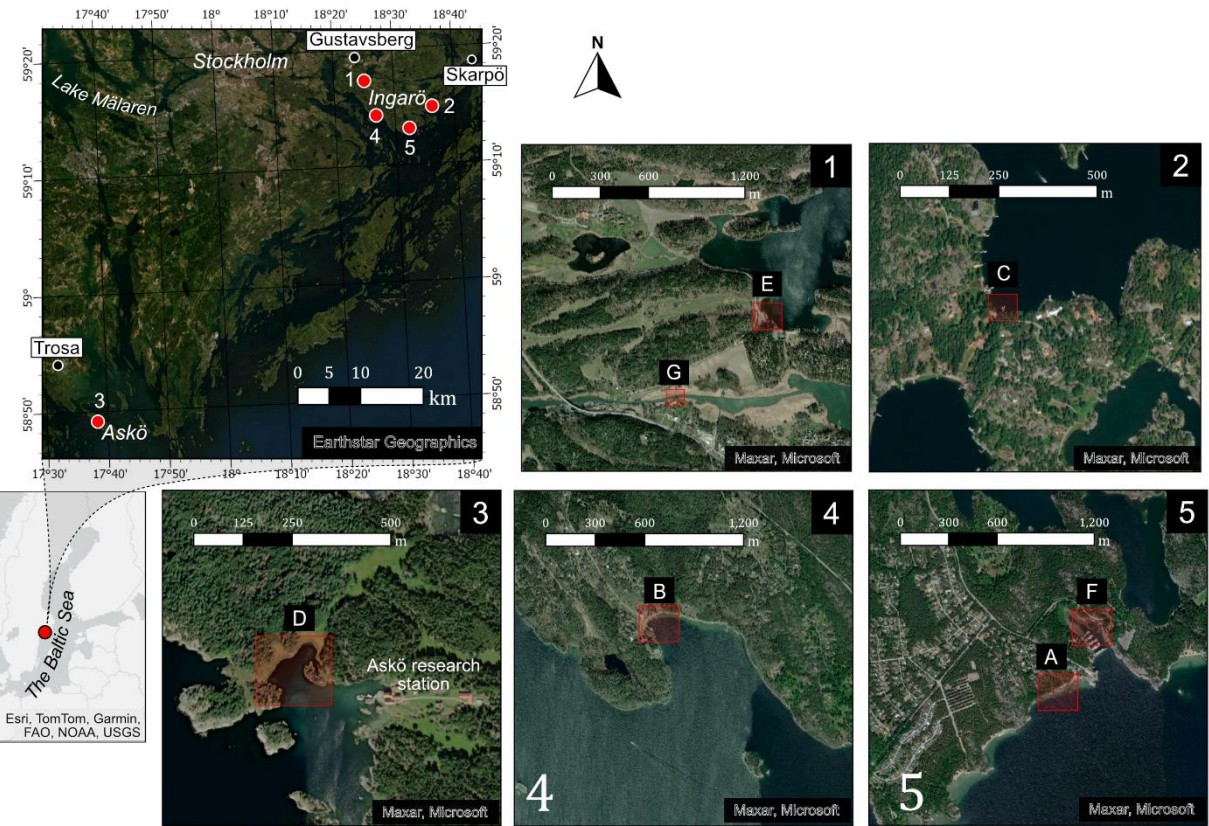

**Figure 1: Map showing the sampling locations (A-F) on Ingarö and Askö. Marked out are also the weather stations in Trosa, Gustavsberg and Skarpö.**

To directly measure $CH_4$ and $CO_2$ fluxes, a floating plastic chamber was connected to an Off-Axis Integrated Cavity Output Spectroscopy (OA-ICOS) Los Gatos Research DLT-100 Greenhouse Gas Analyser (GGA). The chamber design and anchoring function is similar to those used in Schilder et al. (2016), but without an ebullition shield. The round chamber was

covered with aluminium foil to reduce internal solar heating and had a volume of 7500 ml covering a surface area of $0.073 \ m^2$. Foam was attached to the downward-facing sides of the chamber for flotation, and the walls submerged 1.5-2 cm into the water depending on sea state. The chamber was connected to the GGA via two, 20 m long, plastic tubing with an inner diameter of 3 mm. The tubing was connected to the inlet of the GGA via two 100 ml Winkler bottles and an Acro® 50 vent filter to avoid particles and water entering the GGA. The total volume of the chamber, tubing, Winkler bottles, filters, and cavity of the GGA

was ca 7700 ml. Air was circulated using the GGA's internal pump and sampled at a frequency of 20 seconds. The flux was determined from the concentration gradient during the sampling time (see section 'Diffusive flux calculations').



Each flux measurement lasted ~20 minutes, after which the chamber was lifted from the water surface and the GGA allowed to equilibrate with the air. This short measurement time minimises bias from the effect on the flux of the increasing gas concentration within the chamber (Mannich et al., 2019). At each sampling location flux measurements were typically
carried out in transects from close to shore to maximum 30 meters away from shore. The chamber was placed to account for variations in vegetation community and density equally and, where present, also placed over emerged vegetation. Between 3 and 76 flux measurements were done at each location, each sampling period. The sampled area per location was between 40 and 800 m$^2$.

Local wind speed, wind direction, air temperature, atmospheric pressure, and precipitation were measured during flux
sampling using a Eurochron wireless weather station, ECWCC1080, mounted on a mast 1.5 m above the water surface close to the sampling location. Wind speed was adjusted to 10 m height ($U_{10}$) assuming a neutral logarithmic profile (Amorocho & DeVries, 1980). Eq. (1):

$$U_{10} = \frac{U_z}{1 - \frac{\sqrt{C_{10}}}{\kappa}\ln(\frac{10}{z})}$$  (1)

where $U_z$ is the wind speed at the measurement height (m s$^{-1}$), $C_{10}$ is the drag coefficient for shallow water (0.0013), $\kappa$ is the
von Kármán constant (0.4) and z is the measurement height (m).

Regional timeseries of wind speed and air temperatures for Stockholm and Trosa archipelagos were obtained from the Swedish Meteorological and Hydrological Institute (SMHI) from the weather stations in Gustavsberg, Skarpö and Trosa (Fig. 1), wind speed was only available from Skarpö (SMHI, 2024a; SMHI, 2024b).

Water salinity and temperature were measured close to the surface (at ~5 cm depth) with a handheld sensor
(conductivity WTW 340i).

**2.3 Diffusive flux calculations**

The diffusive fluxes of $CO_2$ and $CH_4$ were calculated from the change in gas concentration in the chamber-GGA system, like Eq. (2):

$$F_d = \frac{\Delta C * V}{A * \Delta t}$$  (2)

where $F_d$ is the diffusive flux (mg m$^{-2}$ d$^{-1}$), $\Delta C$ is the gas concentration change measured by the GGA (mg L$^{-1}$), V is the volume of the chamber-GGA system (L), A is the chamber footprint (m$^2$) and $\Delta t$ is the duration of the measurement.

For diffusive flux measurements, the rate of concentration change measured by the GGA is expected to be near-linear. The variations in the rate of change due to changing gas transfer velocity (changes in turbulence in the boundary layer) or variations in the gas concentration during the measurement are expected to be small. This contrasts with ebullition events,
which are apparent in the measured gas concentration as large, sudden changes. To exclude such events and also periods with





other measurement artifacts (e.g., fitting leakage or induced turbulence from chamber deployment), we divided each measurement into five segments (each ~4 minutes long) and performed least-squares linear regression analysis on each segment. Segments that showed an approximate linear change ($R^2 >0.7$) were used for calculating the $\Delta C$.

For a description of the extrapolation of daily fluxes into annual fluxes, see Supplementary Material Text S2 and
Table S1.

### 2.4 Ebullition flux

Ebullition was detected in the measurements as abrupt changes (at least three times the standard deviation of the change from the diffusive flux for a 20 second measurement interval) in concentration detected by the GGA (Supplementary Material Fig. S1). The ebullition events either appeared as a simple, step-up in concentration, or as a peak, followed by a more gradual
increase in concentration characteristic of diffusive exchange afterwards.

Calculating the amount of $CH_4$ released per event was done using Eq. (3):

$$m_e = (\Delta C_e * V) - m_d \tag{3}$$

where $m_e$ is the amount of $CH_4$ released per event (mg), $\Delta C_e$ is the concentration increase during the ebullition event, from that it starts to deviate from the diffusive gradient until the diffusive gradient is restabilized (mg $L^{-1}$), V is the volume of the
chamber-GGA set-up (L) and $m_d$ is the amount of $CH_4$ released via diffusive flux for the same time as the ebullition event lasted (mg) which could be derived from Eq. (2) and the gradient prior to the ebullition event (Supplementary Material Fig. S1).

The daily $CH_4$ emission from ebullition was extrapolated from the $CH_4$ released per event by assuming that the sampled ebullition was representative of the entire location and the entire day. Spatial extrapolation was conducted by
assuming that the number of chamber measurements detecting ebullition for that sampling location and period relative to the total number of chamber measurements for that sampling location and period was proportional to the areal fraction of the bay that released ebullition. The flux per $m^2$ for the period of sampling was then extrapolated over 24 hours to obtain a daily ebullition flux. The extrapolation is summarized in Eq. (4).

$$F_e = m_{e\ tot} * \frac{Ch}{A} * \frac{24}{t} \tag{4}$$

where $F_e$ is the ebullition flux (mg $m^{-2}$ $d^{-1}$), $m_{e\ tot}$ is the total released $CH_4$ by ebullition during sampling in the specific period (mg), Ch is the number of chamber measurements with ebullition (each measurement containing between 1 and 4 ebullition events) divided by the total number of chamber measurements for the specific period, A is the footprint of the chamber ($m^2$), t is the total time that sampling was performed in a location for that month (h).





**2.4 Statistics**

The confidence intervals of the average diffusive flux across different habitats were determined using bootstrapping to resample the flux data (Nelson, 2008). Bootstrapping involves generating multiple new datasets by randomly sampling the observed data with replacement (when a data point is selected from the original dataset to create a new dataset, it is not removed from the pool of possible selections). Each new dataset is the same size as the original dataset and includes repeated values from the original data. By resampling in this way, the method estimates the variability of the sample average without 175 assuming a normal distribution.

The data were first divided into two seasonal groups: 'summer' (May, June, July, and September) and 'winter' (March, October, and December). January was excluded as only two locations were sampled during that period, making the data insufficient for robust analysis. For each habitat and season, the flux data was resampled 2000 times from the original dataset.

Habitat differences in ebullition were evaluated with a Kruskal-Wallis test (Kruskal & Wallis, 1952) which does not assume normal distribution in the data. The Kruskal-Wallis test was complemented with a Bonferroni Multiple Comparison test (Francis & Thunell, 2021), which corrects for type I errors. Differences where $p < 0.05$ was considered to be significant.

**3 Results**

Throughout the text we use the convention that positive fluxes indicate emission of gas from the water surface to the 185 atmosphere, while negative fluxes indicate uptake by the water surface from the atmosphere. Uncertainties in average flux values are reported as the standard error of the mean. Uncertainties in meteorological and environmental data are given as standard deviations.

**3.1 Meteorological and environmental data**

During the sampling periods, locally measured air temperatures ranged from -2.0 to 24.0 °C, with the lowest temperatures 190 recorded in January 2021 and the highest in July 2021. Comparatively, daily mean air temperatures from the SMHI records for the same dates were, on average, 13 % lower than those measured locally (Fig. 2a).

Locally measured $U_{10}$ varied from near zero to 5.8 m s$^{-1}$, with an average of $1.6 \pm 1.1$ m s$^{-1}$ (Fig. 2b). Greater variability in local $U_{10}$ was observed from October to May, whereas measurements during June, July, and September were less variable. However, the monthly-averaged local $U_{10}$ did not exhibit a clear seasonal pattern. When compared to the SMHI wind data for 195 the corresponding dates and times at Skarpö station, the local daily average $U_{10}$ was on average 44 % lower. Over the entire sampling campaign, the local average $U_{10}$ was 47 % lower than the SMHI station's long-term average between 2020 and 2022, which included both day- and nighttime data.



Observed surface water temperatures were between 1.7 and 24.5 °C, with the highest temperatures observed in July, and the lowest in January, similar to the air temperatures. Surface water salinity was between 2.3 and 6.3 ‰ (Table 1), with

>90 % of observed salinities above 4.0. Salinities below 4.0 were measured between January and April, which was coinciding with the period following ice break-up and during spring snowmelt.

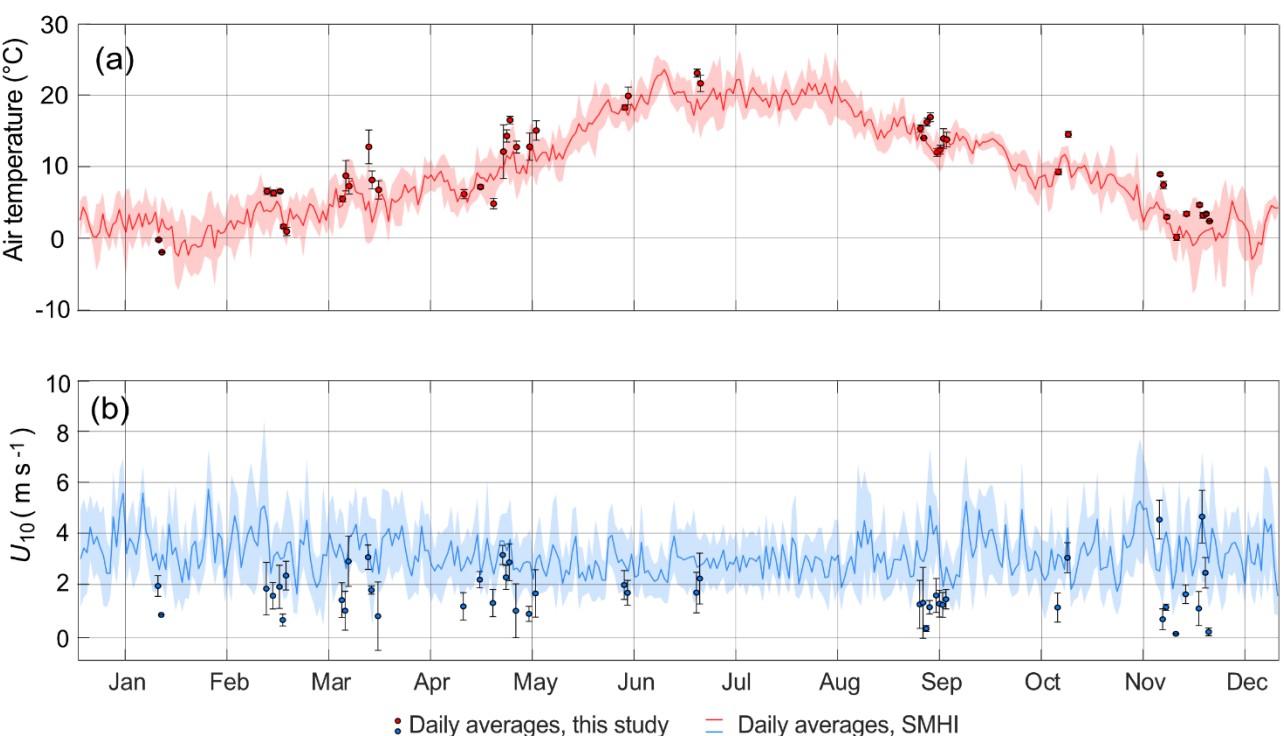

**Figure 2. Daily average air temperatures (a) and wind speeds at 10 m height, $U_{10}$ (b). The local measurements during the flux sampling are shown as circle markers, with whiskers representing standard deviations. The plotted lines are daily average data from SMHI between the years 2020 and 2022. Shaded areas are standard deviations. X-axis tick marks are on the 15th of every month.**

### 3.2 $CO_2$ sea-to-air fluxes

The sea-to-air $CO_2$ fluxes at each of the different sampling locations are shown in Fig. 3a. Omitting January where only two locations were sampled, the monthly average fluxes ranged from -937 ± 161 to 363 ± 87 mg m$^{-2}$ d$^{-1}$ in the macroalgae-covered

coarse sediment habitat (A), from -670 ± 8 to 1 496 ± 68 mg m$^{-2}$ d$^{-1}$ in the sparsely vegetated sand habitat (B), from -673 ± 27 to 1 494 ± 77 mg m$^{-2}$ d$^{-1}$ in the submerged plant-covered sand habitat (C), from -328 ± 17 to 1 725 ± 139 mg m$^{-2}$ d$^{-1}$ in the submerged plant-covered mixed substrate habitats (D and E), and from 294 ± 26 mg m$^{-2}$ d$^{-1}$ to 1 757 ± 95 mg m$^{-2}$ d$^{-1}$ in the reed-covered muds (F and G).



**Figure 3. Boxplots with diffusive sea-air (a) CO₂ and (b) CH₄ fluxes from the chamber measurements, at each location and sampling period. Boxes represents 25ᵗʰ and 75ᵗʰ percentiles, line within the box is the median, whiskers are 100ᵗʰ and 0ᵗʰ percentiles. Note the logarithmic scale in b. Numbers underneath the boxes are number of measurements and letters above are sampling location.**





The macroalgae habitat (A), the submerged plant-covered sand habitat (C) and the mixed substrate habitats showed net $CO_2$ uptake throughout the warmer period of the year between March and September, while the sparsely vegetated sandy habitat (B) only took up $CO_2$ in June and July. The macroalgae habitat (A) was a net annual sink of - $109 \pm 4$ g m$^{-2}$ yr$^{-1}$ since it only emitted small quantities of $CO_2$ in the autumn months of October to December (Table 2, Fig 3a). In the sand and

mixed substrate habitats, the summer uptake was counterbalanced by high emissions in the autumn and winter months, so that these locations were acting as weak sources annually, spanning between $3 \pm 8$ and $111 \pm 16$ g m$^{-2}$ yr$^{-1}$ (Table 2).

The reed-covered muds differed from the other habitats by constantly showing net emission of $CO_2$, except for location F in March. Annually, this habitat was a relatively strong source of $CO_2$ between $227 \pm 23$ and $496 \pm 21$ g m$^{-2}$ yr$^{-1}$ (Table 2).

In January the $CO_2$ emission from the two sampled locations, the submerged plant-covered sand habitat C, and the reed habitat G, were noticeably higher than during adjacent sampling periods of March and November – December. These elevated $CO_2$ fluxes in January corresponded with a time when the other the locations, as well as much of the coastline, were ice covered while these locations were ice free. The monthly average emissions in January in C and G were up to three times higher than those in March and November–December.

**3.3 Diffusive CH$_4$ sea-to-air fluxes**

**Table 2. Yearly fluxes (diffusion and ebullition) of CH$_4$, CO$_2$ and CO$_2$-equivalent flux of both CO$_2$ and CH$_4$ calculated for all locations, given in g m$^{-2}$ yr$^{-1}$. Uncertainties are given as standard errors of the mean. CH$_4$ ebullition fluxes are within brackets**.

| Habitats: | Locations: | CH$_4$ | CO$_2$ | CO$_2$-eq. 20 years | CO$_2$-eq. 100 years |
|---|---|---|---|---|---|
| Coarse sed. – Macroalgae | A | $0.16 \pm 0.01$ | - $109 \pm 4$ | $-94 \pm 5$ | $-102 \pm 4$ |
| Sand - Sparse veg. | B | $3.3 \pm 0.3$ (+5.0) | $74 \pm 13$ | $391 \pm 42$ (+480) | $223 \pm 27$ (+225) |
| Sand – Subm. plants | C | $0.4 \pm 0.03$ (+11) | $14 \pm 12$ | $52 \pm 15$ (+1056) | $32 \pm 13$ (+495) |
| Mixed sub. – Subm. plants | D | $0.7 \pm 0.07$ (+1.6) | $3 \pm 8$ | $70 \pm 15$ (+154) | $35 \pm 11$ (+72) |
| | E | $0.8 \pm 0.06$ (+0.4) | $111 \pm 16$ | $188 \pm 22$ (+38) | $147 \pm 19$ (+18) |
| Muddy sed. – Reeds | F | $1.5 \pm 0.2$ (+0.4) | $227 \pm 23$ | $371 \pm 42$ (+38) | $295 \pm 32$ (+18) |
| | G | $1.0 \pm 0.03$ | $496 \pm 21$ | $592 \pm 24$ | $541 \pm 22$ |





All sampling locations were sources of $CH_4$ throughout the entire annual cycle, consistently showing a flux from the water to the atmosphere (Fig. 3b).

The average monthly diffusive sea-to-air $CH_4$ fluxes varied greatly across habitats. The macroalgae habitat (A) had the lowest flux, ranging from $0.1 \pm 0.01$ to $1.1 \pm 0.1$ mg $m^{-2}$ $d^{-1}$, with no clear seasonal pattern. In contrast, the submerged plant-covered sand habitat (C) exhibited slightly higher fluxes, ranging from $0.2 \pm 0.01$ to $3.3 \pm 0.3$ mg $m^{-2}$ $d^{-1}$, and displayed a clear seasonal trend, with elevated fluxes during warmer months. The mixed substrate habitats (D and E) had fluxes ranging from $0.7 \pm 0.1$ to $4.6 \pm 0.7$ mg $m^{-2}$ $d^{-1}$, and similar to sand habitat C the fluxes were higher in summer. The reed habitats (F and G) also showed increased fluxes in summer, but with a wider range from $0.5 \pm 0.1$ to $15 \pm 3.6$ mg $m^{-2}$ $d^{-1}$. The sparsely vegetated

sand habitat (B) had the highest fluxes overall, with monthly averages ranging from $2.3 \pm 0.3$ to $26 \pm 11$ mg $m^{-2}$ $d^{-1}$. Similar to the macroalgae habitat (A), this habitat demonstrated little seasonal dependence, with the highest fluxes recorded during November–December.

Annually, the diffusive $CH_4$ fluxes ranged from $0.16 \pm 0.01$ g $m^{-2}$ $yr^{-1}$ in the macroalgae habitat (A) to $3.3 \pm 0.3$ g $m^{-2}$ $yr^{-1}$ in the sparsely vegetated sand habitat (B) (Table 2).

As observed for $CO_2$, the diffusive $CH_4$ fluxes from the ice-free locations in January were unusually high. Fluxes from the sand habitat C ($0.7 \pm 0.02$ mg $m^{-2}$ $d^{-1}$) were an order of magnitude higher, while fluxes from the reed habitat G ($4.0 \pm 0.3$ mg $m^{-2}$ $d^{-1}$) were approximately three times greater than the respective fluxes measured at these locations during November–December.

### 3.4 $CH_4$ ebullition

$CH_4$ ebullition was detected during measurements in all habitats except the macroalgae habitat (A) (Fig. 4) and while ebullition was measured from March to October, most events were observed between July and September (Fig. 4c).

During the ebullition events, the amount of $CH_4$ released per bubble ranged from 0.05 to 2 688 μg (Fig. 4a), with both the lowest and highest values recorded in the submerged plant-covered sand habitat (C). The sparsely vegetated sand habitat (B) released between 332 to 1 455 μg per bubble, which was significantly higher than that observed in the mixed substrate

habitats (D and E) (1.2–679 μg) and the reed habitat (F) (36–151 μg).

At the locations and during the months when ebullition was detected, between 4–40 % of the chamber experiments recorded ebullition. Extrapolating the ebullition over a 24h period yielded $CH_4$ ebullition fluxes between 0.001 and 232 mg $m^{-2}$ $d^{-1}$ (Fig. 4b). The sparsely vegetated sand habitat (B) had ebullition fluxes between 6 and 83 mg $m^{-2}$ $d^{-1}$, the submerged plant-covered sand habitat (C) had the largest range from 0.001 to 232 mg $m^{-2}$ $d^{-1}$, the mixed substrate habitats (D and E) ranged

between 0.5 and 83 mg $m^{-2}$ $d^{-1}$, and the reed habitat (G) had only one ebullition flux recorded in September, 12 mg $m^{-2}$ $d^{-1}$. No statistically significant differences could be found between the habitats, possibly due to the low number of fluxes, but generally the sand habitats showed the highest fluxes.



For the locations that contained bubbles, between 0.3 and 96 % of the total (ebullition + diffusion) CH$_4$ flux could be attributed to ebullition for a specific sampling month. When extrapolating both fluxes over a year, assuming that ebullition was absent the months where the measurements did not detect it, ebullition flux accounted for between 21 and 98 % of the total annual flux (Table 2).

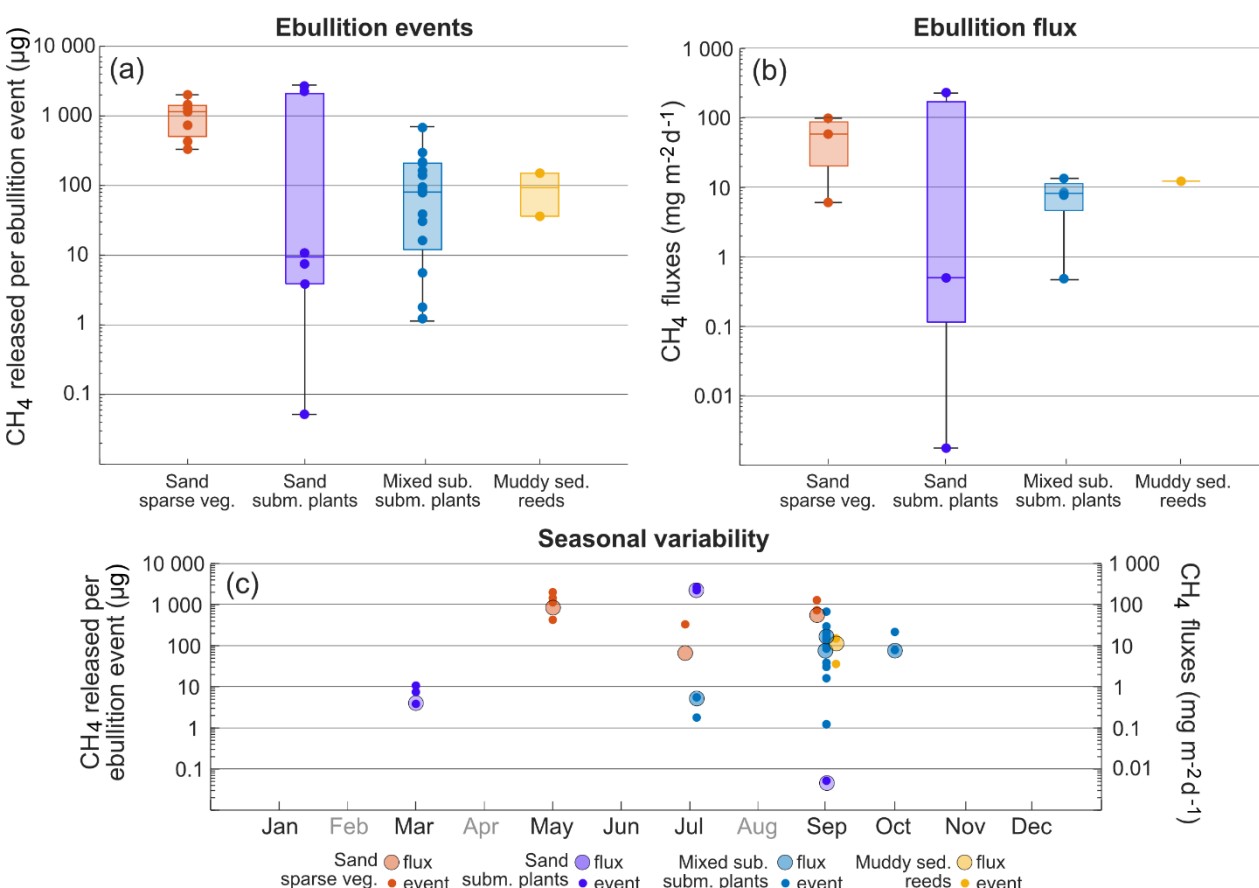

**Figure 4. Boxplots with CH$_4$ released per ebulliton event (a) and CH$_4$ ebullition flux calculated for a specific location and period (b). Boxes represents 25$^{th}$ and 75$^{th}$ percentiles, line within the box is the median, whiskers are 100$^{th}$ and 0$^{th}$ percentiles. CH$_4$ released per ebulliton event and CH$_4$ ebullition flux is also plotted for the specific months where it was detected (c). Months that has not been sampled are in grey.**




## 3.5 Habitat differences and variability

The bootstrapping (Fig. 5) determined that the summer average $CO_2$ flux (May to September) in the macroalgae habitat (A)

had a 95 % confidence interval of -720 to -481 mg m$^{-2}$ d$^{-1}$, while the reed habitats (F and G) had an average between 575 and

1 011 mg m$^{-2}$ d$^{-1}$. These two habitats stood out as being statistically distinct at the 95 % level from any of the other habitats in

summer (Fig. 5a). In contrast, the sparsely vegetated sand habitat (B), with a summer flux confidence interval of -386 to -48

mg m$^{-2}$ d$^{-1}$ could not be differentiated from either the submerged plant-covered sand habitat (C) (-389 to -258 mg m$^{-2}$ d$^{-1}$) or

the mixed substrate habitats (D and E) (-202 to -36 mg m$^{-2}$ d$^{-1}$). However, the latter two habitats did exhibit statistical

270 differences from each other.

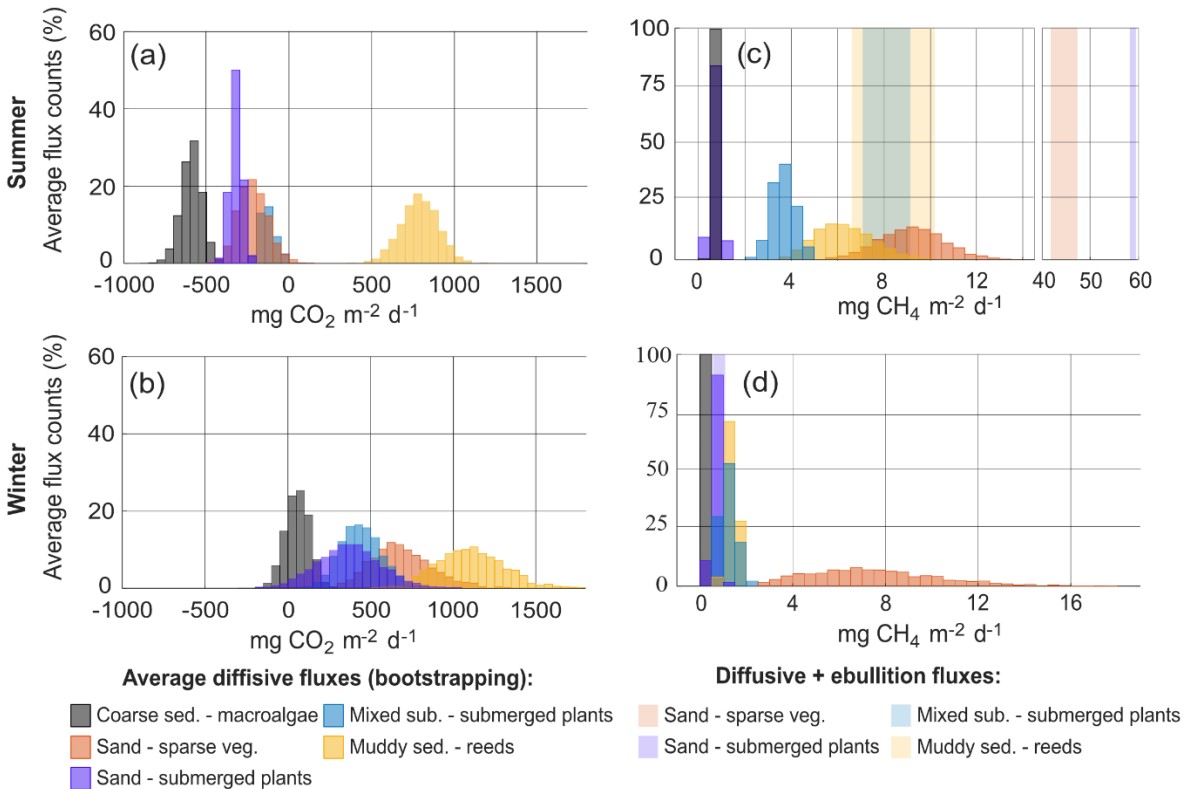

**Figure 5. Average diffusive flux values obtained from the bootstrapping exercise for $CO_2$ (a and b) and $CH_4$ (c and d), divided up into the summer months May, June, July, and September (a and c) and winter months March, October, and November-December (b and d). January has been excluded since only two locations were sampled. The 95% confidence interval of the diffusive $CH_4$ has been added together with the average ebullition flux and are displayed as pale blocks in c and d.**



During the ice-free months during winter (October, November, December, and March) none of the habitats had a distinct average $CO_2$ flux and each habitat overlapped at least one other habitat (Fig. 5b), although visual examination of the distributions of the average flux values indicates an increase from the macroalgae habitat (A) (-90 to 191 mg m$^{-2}$ d$^{-1}$) followed by, in order of increasing flux, the submerged plant-covered sand habitat (C) (-91 to 631 mg m$^{-2}$ d$^{-1}$), the mixed substrate habitats (D and E) (216 to 691 mg m$^{-2}$ d$^{-1}$), the sparsely vegetated sand habitat (B) (375 to 1 040 mg m$^{-2}$ d$^{-1}$) and lastly the reed habitats (F and G) (677 to 1 327 mg m$^{-2}$ d$^{-1}$).

In summer, the average diffusive $CH_4$ flux in the macroalgae habitat (A) (0.5 to 0.9 mg m$^{-2}$ d$^{-1}$) could not be distinguished from the submerged plant-covered sand habitat (C) (0.4 to 1.1 mg m$^{-2}$ d$^{-1}$), although both these habitats were statistically distinct from the other habitats (Fig. 5c). However, including ebullition for the two sand habitats (B and C) indicate an average total (diffusive + ebullition) summer flux of around 45 and 60 mg m$^{-2}$ d$^{-1}$, respectively, clearly separating these two habitats from the rest. The mixed substrate habitats (E and D) and the reed habitats (G and F) could not be distinguished from each other in summer when both ebullition and diffusion were considered, and both had an average flux centred around 8 mg m$^{-2}$ d$^{-1}$.

In winter, all habitats except the sparsely vegetated sand habitat (B) had confidence intervals of their average flux under 2 mg m$^{-2}$ d$^{-1}$, while the sand habitat (B) had a confidence interval between 3.1 and 14 mg m$^{-2}$ d$^{-1}$ (Fig. 5d).

## 3.6 $CO_2$-equivalent fluxes

To calculate the $CO_2$-equivalent flux for $CH_4$ over a 20- and 100-year time period, the flux was multiplied by 96 or 45, respectively, to account for the sustained global warming potential of $CH_4$ (Neubauer & Megonigal, 2015).

Considering both a 20- and a 100-year time scale, the $CO_2$-equivalent flux of $CH_4$ accounted for between 83 and 99 % of the total radiative forcing added to the atmosphere from the sand habitats on an annual basis (B and C), between 33 and 99 % from the mixed substrate habitats (D and E) and between 8 and 45 % in the reed habitats (F and G) (Table 2). In the macroalgae habitat (A) the $CO_2$-equivalent flux of $CH_4$ offset the annual $CO_2$ uptake with between 7–14 %.

On a monthly basis, the $CO_2$-equivalent fluxes of $CH_4$ had the greatest impact in the summer months due to the contributions from ebullition (Fig. 6).



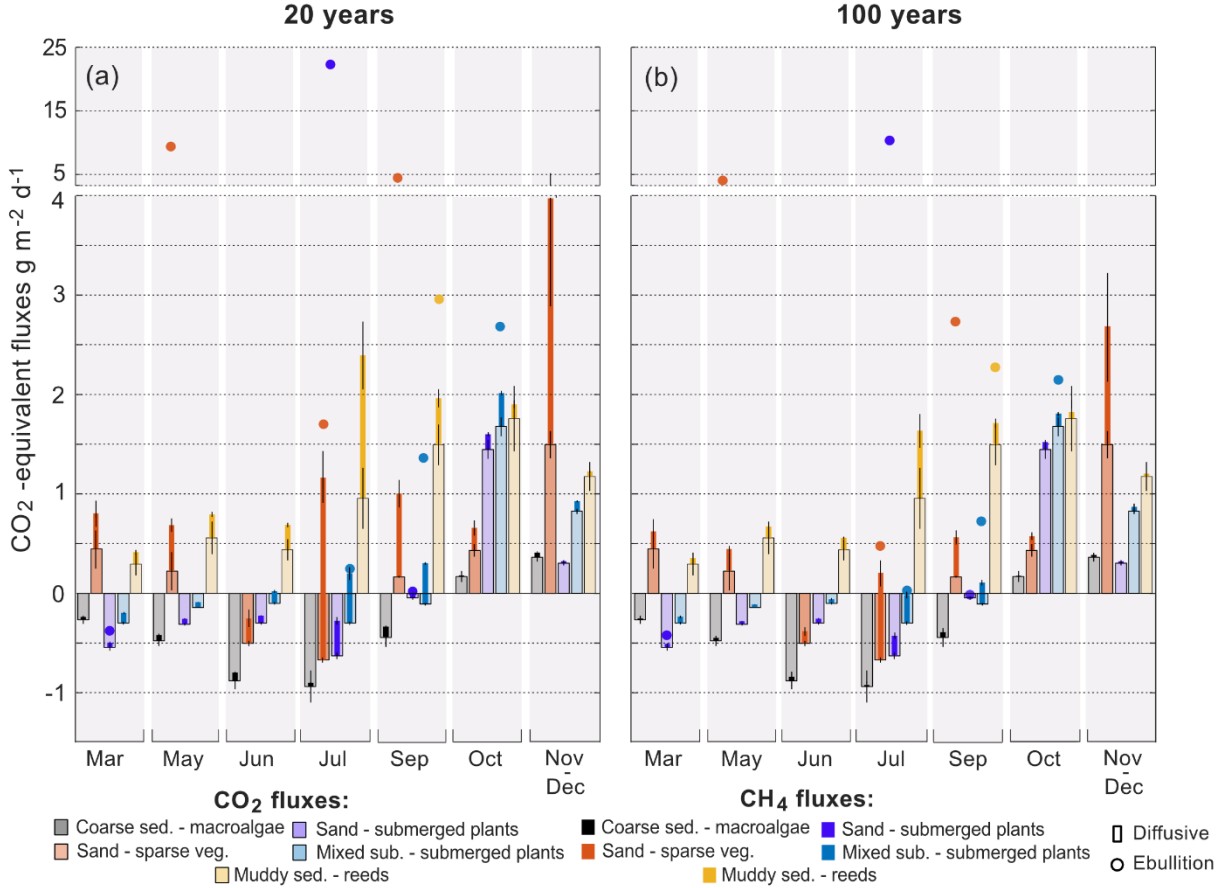

**Figure 6. Diffusive CO₂ and CH₄ as average habitat CO₂-equivalent fluxes for each sampling period, except January when only two locations were sampled, for both a 20-year time period (a) and 100-year time period (b). The CO₂-equivalent CH₄ ebullition fluxes are not averages for each habitat but plotted as the calculated values for each location containing ebullition. Note the broken y-axis. CH₄ fluxes (diffusive + ebullition) are stacked on the CO₂ fluxes and therefore represent the flux value of both gases. Error bars are the standard error of the mean.**

## 4 Discussion

### 4.1 Sediment and vegetation type as predictor for flux

The shallow-water coastal zone demonstrates significant variability in CO₂ and CH₄ fluxes, spanning over two orders
of magnitude over short spatial (< 50 m) and temporal (< 24 h) scales (Asmala & Scheinin, 2023; Roth et al., 2023). This





variability complicates efforts for quantification and upscaling of the coastal contribution of these greenhouse gases. To address this, we assessed whether these fluxes could be reliably constrained within habitat types already defined within HELCOM HUB (HELCOM, 2013). Linking flux variability to habitat types offers a potential pathway to streamline upscaling efforts.


**Table 3. Sea-to-air $CO_2$ and/or $CH_4$ (diffusive + ebullition) fluxes in shallow, near-shore waters (mean depth <4 m) in the Baltic Sea, where the study periods have stretched at least a few months. Fluxes are reported as the range of monthly averages, if not denoted with a \*, in which case it is an average over the whole study period. Fluxes are in the unit of mg m$^{-2}$ d$^{-1}$.**

| Habitat | | $CH_4$ fluxes | $CO_2$ fluxes | Months | Reference |
|---|---|---|---|---|---|
| Rocks and boulders | Macroalgae | 0.1 – 1.1 | -385 – 117 | Year-round | (Roth et al., 2023) |
| Coarse sediments | No vegetation | 0.7 – 2 | | Jun. – Oct. | (Lundevall-Zara et al., 2021) |
| | Macroalgae | 0.1 – 1.8 | -937 – 363 | Year-round | This study (A) |
| | Submerged rooted plants | 0.1 – 2.9 | -763 – 390 | Year-round | (Roth et al., 2023) |
| Sand | Sparse or no vegetation | 0.4 – 10 | | Jun. – Oct. | Lundevall-Zara et al. (2021) |
| | | 2.3 – 88 | -670 – 1496 | Year-round | This study (B) |
| | Submerged rooted plants | 0.2 – 235 | -673 – 1494 | Year-round | This study (C) |
| Muddy sediments | No vegetation | 0.1 – 2.5 | -132 – 326 | Year-round | (Roth et al., 2023) |
| | Phragmites australis | 0.5 – 15 | -62 – 3512 | Year-round | This study (G and F) |
| | | 206\* – 297\* | | Apr. – Nov. | (Koch et al., 2014; Liikanen et al., 2009) |
| | Mixed epibenthic biotic structures | 0.3 – 163 | | Jun. – Oct. | (Lundevall-Zara et al., 2021) |
| Mixed sediments | Submerged rooted plants | 0.7 – 16 | -328 – 1725 | Year-round | This study (D and E) |

Coarse sediment habitats with macroalgae demonstrated low $CO_2$ and $CH_4$ flux variability and a total absence of $CH_4$ ebullition, likely reflecting limited organic matter deposition and constrained sediment respiration and methanogenesis (Chen et al., 2022). This habitat exhibited annual net $CO_2$ uptake agreeing with prior studies of similar environments, including

macroalgae on rocks and boulders and submerged rooted plants on coarse sediment (Roth et al., 2023) and the $CH_4$ fluxes



align closely with other coarse sediment studies in the Baltic Sea, differing by less than 1.1 mg m$^{-2}$ d$^{-1}$ (Lundevall-Zara et al., 2021; Roth et al., 2023) (Table 3). This consistency positions coarse sediments as robust reference categories for regional $CO_2$ and $CH_4$ flux scaling. Moreover, their similarity to rock and boulder habitats suggests potential for broader classification schemes.

Reed-dominated muddy habitats displayed greater variability in $CO_2$ fluxes. While brackish wetlands with Phragmites vegetation are typically reported as $CO_2$ sinks in summer outside the Baltic Sea (Martin & Moseman-Valtierra, 2015; Sanders-DeMott et al., 2022), this study found predominantly net emissions during summer. Persistent $CO_2$ oversaturation across seasons distinguishes the habitat in this study and likely reflects high remineralization rates of either allochthonous or autochthonous carbon (Chen et al., 2022). While this habitat type is less consistent with the existing literature, the clear
deviation from the other habitats in this study underscores the need for a separate categorization.

The sand and mixed substrate habitats, despite differing submerged plant compositions, exhibited overlapping $CO_2$ flux ranges and similar seasonal dynamics, indicating broadly comparable biogeochemical characteristics. While differences in primary production, respiration, and carbon burial, linked to the vegetation community, may exist and could be responsible for a part of the variability in flux magnitudes or temporal patterns (Gattuso et al., 1998), these differences appear insufficient
to separate the habitats in terms of the average flux. Consequently, for large-scale flux estimations or upscaling efforts, detailed knowledge of subtle variations in vegetation community composition or sediment grain size may be less critical for the $CO_2$ flux. Instead, focusing on dominant habitat features, such as overall sediment type and general vegetation presence, could provide a practical and reliable framework for flux prediction for these habitat groups.

$CH_4$ winter fluxes were consistently low across habitats, aligning with the temperature sensitivity of methanogenesis
(Yvon-Durocher et al., 2014). Habitat differences appeared less influential during this season, though the sparsely vegetated sand habitat exhibited anomalously elevated and variable fluxes with a 10 mg m$^{-2}$ d$^{-1}$ confidence interval of the average winter flux. This outlier may reflect unaccounted drivers or localized factors not described by the HELCOM classification.

In summer, $CH_4$ flux variability increased drastically, mainly due to ebullition which amplified total fluxes by up to an order of magnitude. Our findings suggest that the variability in diffusive $CH_4$ fluxes may be of secondary importance when
compared to the significant contribution of sea-air ebullition. Bubble emission not only overshadowed diffusive flux variability but also defined the upper limit of the flux magnitude in many habitats, similar to previous findings in the Baltic Sea shallow-water zone (Lundevall-Zara et al., 2021). The upper limits of $CH_4$ fluxes in sand, mixed substrate and muddy sediment habitats are variable between the studies performed in the Baltic Sea (Table 3), but are up to two orders of magnitude higher for the studies using floating chambers which can incorporate the bubble flux (Liikanen et al., 2009; Koch et al., 2014; Lundevall-
Zara et al., 2021) than for the studies using concentration gradients between the water and air and then calculate a diffusive flux (Roth et al., 2023).

Contradicting previous findings that the majority of ebullition comes from muddy, organic-rich sediments (Crawford et al., 2014; Lundevall-Zara et al., 2021), this study identified the sand habitats as that most prone to ebullition. While it cannot be overruled that organic-rich sediment underlies the sand and contributes to the ebullition, the finding shows that assumptions



that widespread ebullition is isolated to a habitat that would be classified as a muddy sediment habitat will underestimate the
area of the coastline that exhibits intense $CH_4$ ebullition.

## 4.2 Constraining the ebullition flux

The ebullition data from this study not only corroborates previous findings that $CH_4$ ebullition dominates coastal $CH_4$ emissions
(Weber et al., 2019), but further demonstrates that ebullition could be the dominant component of the total coastal carbon-
based greenhouse gas flux. Most of the previous estimates for coastal $CO_2$ and $CH_4$ fluxes in the Baltic Sea have relied on bulk
methods based on sea-air concentration gradients and *k* parameterization (e.g. Ma et al., 2020; Roth et al., 2022; Asmala &
Scheinin, 2023). While effective for assessing diffusive exchange, and hence also for the sediment-derived gas ebullition that
subsequently dissolves in the water column (Hermans et al., 2024), these approaches fail to account for direct bubble-mediated
transport across the water-air interface, thereby leading to systematic underestimation of coastal $CH_4$ fluxes.

A common characteristic of ebullition studies in near-shore waters, including this one, is the large range of the
observed fluxes which typically vary by over four orders of magnitude (Lundevall-Zara et al., 2021; Wang et al., 2021;
Żygadłowska et al., 2024). However, our findings indicate that habitat type provides a useful predictor of both the frequency
of ebullition events and, to a lesser extent, the amount of $CH_4$ released per event. This suggests that habitat-based classifications
could serve as a valuable tool for constraining ebullition flux estimates, offering a more structured framework for quantifying
$CH_4$ emissions in coastal ecosystems.

A key challenge remains in scaling episodic ebullition events into time- and space-averaged fluxes. Variability in
bubble size, release frequency, and lateral distribution introduces significant uncertainty in determining representative flux
values. Consequently, the methodological approach used to integrate ebullition events into broader estimates can significantly
influence reported fluxes. Nevertheless, our results align with previous observations from a eutrophic coastal basin of the North
Sea (Żygadłowska et al., 2024), and from the island of Askö in the Baltic Sea (Lundevall-Zara et al., 2021). Reported ebullition
fluxes in the North Sea study were up to 3920 mg m$^{-2}$ d$^{-1}$, while at Askö the average monthly ebullition and diffusive fluxes
were up to 163 mg m$^{-2}$ d$^{-1}$. The maximum monthly diffusive and ebullition flux observed in this study (235 mg m$^{-2}$ d$^{-1}$) falls
within the same range, despite differences in methods used to integrate ebullition into time-averaged estimates.

## 4.3 The importance of winter fluxes for the annual budget

In January, the emission fluxes of both $CO_2$ and $CH_4$ at ice-free locations were high despite the coldest water temperatures of
the sampling campaign, when both methanogenesis and respiration are expected to be lowest (Thamdrup et al., 1998; Yvon-
Durocher et al., 2014). Furthermore, since the solubility of $CO_2$ and $CH_4$ increases with decreasing water temperature, it should
result in less outgassing to the atmosphere (Lucile et al., 2012; Guo & Rodger, 2013).

Previous studies have found increased gas fluxes following ice breakup in aquatic environments and suggested it to
be a period of significant contribution to the annual flux budget (Jansen et al., 2019; Wang et al., 2023). During ice-covered





periods, gases accumulate under the ice, with little-to-no sea-air exchange and limited oxidation of $CH_4$ due to the low temperatures, which are later allowed to be outgassed during ice breakup (Denfeld et al., 2018; Roth et al., 2022). In our study, no active ice melt or break up was observed at the time of sampling, and ice observations were not made prior to the sampling periods. We speculate that the higher fluxes may be due to earlier ice breakup, or horizontal transport of water from ice-covered
areas.

Assuming that this elevated flux would persist for half a month to a full month (depending on whether it originates from previous local ice breakup or from transport from other ice-covered areas), its contribution to the annual exchange of $CO_2$ would range between 10 and 22 %, and to the annual diffusive emission of $CH_4$, between 3 and 12 %. However, from the perspective of the coastal greenhouse gas budget, these emissions may simply represent a shift in the timing or location of
release rather than an overall increase in total flux. Establishing baseline winter emissions of $CO_2$ and $CH_4$ during ice-free conditions may suffice for budgetary assessments, as the presence of ice would primarily redistribute these emissions across time and space rather than add to the annual total.

## 4.4 Coastal $CO_2$ and $CH_4$ budget of the Stockholm archipelago

Given the diversity of habitats along the Baltic Sea coastline and the relatively high anthropogenic pressure on the Stockholm
archipelago (Hill & Wallström, 2008; Kautsky, 2008), extrapolation beyond this region could introduce large uncertainties. The Stockholm archipelago, which absorbs much of the city's anthropogenic load, differs from other urban coastal areas in the Baltic Sea, where habitat-specific flux ranges must be assessed independently. However, despite this regional specificity, the archipelago holds roughly one-fourth of Sweden's total coastline (mainland and islands) (Statistics Sweden, 2020), making it representative of a significant portion of the Swedish near-shore shallow-water zone.

The agreement of the temperature and wind data between the SMHI record (SMHI, 2024a, 2024b) and the local measurements suggests that the dataset has captured the seasonal cycle in the area, with deviations likely arising from differences in geographical settings of the SMHI stations and the sampling locations, as well as shorter averaging periods of the local measurements. However, we note that sampling was limited to wind speeds below 5.8 m s$^{-1}$. Due to the commonly suggested non-linear dependence of gas transfer on wind speed (Wanninkhof et al., 2009), infrequent high wind periods may
disproportionately influence the average flux.

Another source of uncertainty is the absence of nighttime flux data. While diurnal variations in $CH_4$ fluxes have been observed in coastal waters of the Baltic Sea (Roth et al., 2022) and on Sweden's west coast (Henriksson et al., 2024), daytime increases were inconsistent across habitats and months. Further, reported $CH_4$ variability generally falls within the diffusive bootstrapping confidence interval for summertime fluxes in this study (0.02 to 6 mg m$^{-2}$ d$^{-1}$), making it a minor uncertainty
compared to the ebullition flux. For $CO_2$ fluxes, photosynthesis is expected to peak in the summer afternoons which may have led to a slight overestimation of uptake (Roth et al., 2023), though measured summer $CO_2$ fluxes in this study do align with prior estimates for coastal habitats which include nighttime values (Honkanen et al., 2024).



Despite these uncertainties, the dataset can provide a first estimate of the coastal shallow-water sea-air gas budget for the Stockholm area. Mattisson (2005) mapped the Stockholm archipelago based on the EUNIS classification scheme (Davies et al., 2004), a habitat classification scheme developed for Europe, which the HELCOM HUB is compatible with, only the latter is specifically developed for the Baltic Sea (HELCOM, 2013). The mapping found that the most common habitat groups in shallow waters (<6 m) were bedrock or coarse sediments, which take up 40 % of the area (total mapped area: 681 km$^2$). Applying the annual flux of the coarse sediment with macroalgae habitat to this area yields fluxes of -30 Gg $CO_2$ yr$^{-1}$ and 0.04 Gg $CH_4$ yr$^{-1}$. Reed beds account for 1.6 % of the total area, and muddy sediments without specified vegetation account for another 18 %, applying the fluxes from the muddy sediment with reeds habitats yield a flux of between 30 and 67 Gg $CO_2$ yr$^{-1}$ and 0.01 to 0.03 Gg $CH_4$ yr$^{-1}$. Sublittoral sand, with or without vegetation (not distinguished in Mattisson (2005)), covers 2 % of the shallow water area. Applying the lowest and highest flux estimate for the sand habitat yields fluxes ranging between 0.04 and 1.7 Gg $CO_2$ yr$^{-1}$ and 0.1 and 0.2 Gg $CH_4$ yr$^{-1}$. The remaining 38 % is described as a mixture of consolidated clay, bedrock, and more mobile sediments, but is not described in terms of its vegetation. Applying the highest and lowest values from the annual fluxes from the coarse sediment habitat, the sand habitats and the mixed substrate habitats yields a large possible range of -28 to 29 Gg $CO_2$ yr$^{-1}$ and 0.04 to 2.9 Gg $CH_4$ yr$^{-1}$.

Combining both fluxes, the $CO_2$-equivalent flux of the shallow-water coastal zone in the Stockholm archipelago adds up to between 0.005 and 0.4 Tg $CO_2$-eq yr$^{-1}$ or -0.01 and 0.2 Tg $CO_2$-eq yr$^{-1}$ over a 20- or 100-year time period, respectively. At a regional scale, the Stockholm urban area emits approximately 1.2 Tg $CO_2$-eq yr$^{-1}$ (100-year timescale) from the energy and transport sectors alone (Stockholms stad, 2024). While the coastal zone's emissions are small by comparison, its contribution to the regional carbon budget still has the potential to offset efforts to reduce overall emissions.

**4.5 Implications for the coastal zone as a tool in climate mitigation**

Distinguishing between air-sea fluxes and sediment carbon sequestration is crucial for accurately determining and communicating the climate mitigation potential of a habitat (Johannessen & Christian, 2023). While most of the sampling locations show a net efflux of $CO_2$ to the atmosphere, they could still sequester organic carbon. An aquatic system can act as both a sink for organic carbon in the sediments and at the same time show sea-air $CO_2$ and $CH_4$ emissions to the atmosphere, due to the addition of allochthonous carbon that subsidises habitat respiration, indicating that these two properties are not always linked (Santoso et al., 2017). In fact, the coarse sediment with macroalgae habitat that showed the highest net uptake of $CO_2$ most likely has the lowest sediment carbon sequestration due to the exposed setting, limiting deposition. While it is true that carbon buried in anoxic sediments can be stored and isolated from atmospheric exchange for 1000+ years (Dahl et al., 2024), it is the actual exchange at the sea-air interface that directly affects atmospheric concentrations (Van Dam et al., 2021).



Integrating the carbon burial in coastal sediments into Nationally Determined Contributions (Herr & Landis, 2016)
or in carbon trading (Claes et al., 2022) as a means of offsetting fossil fuel $CO_2$ emissions, without simultaneously accounting
for the coastal sea-air exchange, risks undermining national and global climate targets (Williamson & Gattuso, 2022).
Including the coastal zone in national carbon budgets is therefore more complex than the terrestrial environment and should
in all cases include simultaneous measurements of $CO_2$ and $CH_4$ over the sea-air boundary layer. As evident from the results
in this study, $CH_4$ can play a substantial, if not dominant, role in this exchange and should be assessed with a method that can
account for the ebullition component of the flux as well.

**5 Conclusions**

The study quantified seasonal and annual fluxes of $CO_2$ and $CH_4$ from five distinct coastal, shallow water habitat
types (macroalgae-covered coarse sediments, sparsely or densely vegetated sands, submerged plant-covered mixed substrates,
and reed-dominated muds) in the archipelago surrounding Stockholm and Trosa along the central Baltic Sea coast. Significant
differences between the flux distributions in the various habitats were found for both diffusive $CO_2$ and $CH_4$ fluxes, primarily
in summertime. Macroalgae-covered coarse sediments and reed-dominated muds stood out as having distinct $CO_2$ flux ranges,
whereas the other three habitats showed more similarities. The occurrence of $CH_4$ ebullition and variation in flux magnitude
allowed for a clear distinction between the habitats where the sand habitats had the highest emissions, followed by the reeds
and mixed substrate habitats and lastly, the macroalgae habitat. The annual net $CO_2$-equivalent exchange indicated that four
out of five habitats were net sources of carbon-based greenhouse gases to the atmosphere where $CH_4$ ebullition dominated the
exchange in three out of five habitats. An initial estimate of the budget of carbon-based greenhouse gases in the shallow-water
coastal zone of the Stockholm archipelago suggests that this area is likely a weak source of carbon-based greenhouse gases to
the atmosphere, with a range of 0.005 to 0.4 Tg $CO_2$-eq $yr^{-1}$, or -0.01 to 0.2 Tg $CO_2$-eq $yr^{-1}$ over a 20- and 100-year time scale
respectively.

**Data availability statement**

Data and metadata that support the findings of this manuscript are available at the Bolin Centre Database and can be
revised at: (the data will be published openly and receive a DOI upon publication of the manuscript). For reviewers a link to
view the data before publication has been provided.

**Author contribution statement**

TB was responsible for data curation, formal analysis, visualization, and writing of the original draft. JP and VB were
responsible for providing resources, supervision, and writing – review and editing. VB was responsible for conceptualization.
All authors contributed to the investigation.



**Competing interests**

The authors declare that they have no competing interests.

**Acknowledgments**

Thea Bisander acknowledge support from the School of Natural Sciences, Technology and Environmental Studies at Södertörn University. Volker Brüchert and John Prytherch acknowledge support from the Department of Geological Sciences and the Department of Meteorology at Stockholm University. The authors would like to express gratitude to M. Krisch Svärdby for assistance with fieldwork and to the staff at the Askö Laboratory for help with logistics when fieldwork was carried out at 480 the station.

**Financial support**

Funding for field work was provided by the Bolin Centre for Climate Research at Stockholm University in the research area Biogeochemical Cycles and Climate Change through the project Scaling Trace Gas Exchange in Coastal 485 Environments to VB.

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
