# Peer review of "Methane ebullition as the dominant pathway for carbon sea-air exchange in coastal, shallow water habitats of the Baltic Sea"

_EGUsphere, 2025_

## Author Comment (AC2)

Supplementary material for

[revised manuscript text omitted]

**Text S3. Extended methods: Detection of ebullition**

Ebullition events were detected as concentration spikes in the flux measurements. These spikes were similar to those seen recorded by the GGA when the analyser was run in the lab, in a closed loop (inlet and outlet tubes were connected to each other) and three injections of 50 ml 10 ppm $CH_4$ standard gas was introduced to the system (Fig. S1a).

Diffusive flux curves measured with the GGA in the field and approved for flux calculations were clean in appearance, lacking peaks and drastic (at least three times the standard deviation of the change from the diffusive flux for a 20 second measurement interval) gradient changes (Fig. S1b). Flux measurements where disturbances by the chamber, leakage in the tubing, or other interference could not be ruled out was omitted from further analysis, such an example can be seen in Fig. S1f where spikes appear in both $CO_2$ and $CH_4$ concentrations and the surrounding diffusive flux curves are noisy.

The measurement was interpreted to contain ebullition when the previous diffusive flux curves had a steady rate of change, $CO_2$ was relatively unaffected at the times of $CH_4$ peaks and no other interference, as mentioned above, could be interfered (Fig. S1c-e). These curves could either contain peaks of $CH_4$ (Fig. S1c-d) or a step-up in $CH_4$ concentration (Fig. S1e), these were both further analysed to obtain the ebullition flux.

**Table S1.** R-square values for linear least-squared relationships between diffusive $CH_4$ or $CO_2$ or ebullitive $CH_4$ (marked in cursive) and the environmental parameters wind speed ($U_{10}$), water temperature, water depth and distance to shore. For relationships between $U_{10}$ and $CO_2$, absolute $CO_2$ values have been used. Bold R-squares indicate statistically significant ($p < 0.05$) relationships. Negative relationships are marked with (-). The relationships with ebullitive flux are only plotted for the locations where sufficient ebullition events to do regression analysis has occurred ($n > 3$), and on the fluxes calculated for single ebullition events.

| Habitats | Locations | $U_{10}$ | | Water temp | | Water depth | | Distance to shore | | Salinity | |
|---|---|---|---|---|---|---|---|---|---|---|---|
| | | $CH_4$ | $CO_2$ | $CH_4$ | $CO_2$ | $CH_4$ | $CO_2$ | $CH_4$ | $CO_2$ | $CH_4$ | $CO_2$ |
| All | | **0.02** | **0.03** | **0.13** | **0.03 (-)** | 0.00 | 0.00 | 0.01 | **0.04 (-)** | 0.00 | **0.18 (-)** |
| | | *0.06* | | *0.01* | | ***0.15*** | | *0.10* | | *0.05* | |
| Coarse sed. - Macroalgae | A | **0.60** | 0.09 | **0.13** | **0.33 (-)** | 0.01 | 0.00 | 0.03 | 0.06 | 0.01 | 0.00 |
| Sand – sparse veg. | B | **0.34** | **0.33** | **0.08** | **0.45 (-)** | 0.04 | 0.04 | 0.00 | 0.02 | 0.00 | 0.04 |
| | | *0.16 (-)* | | *0.49 (-)* | | *0.11* | | *0.06 (-)* | | *0.09* | |
| Sand – subm. plants | C | **0.37** | **0.23** | **0.16** | 0.02 | 0.00 | 0.03 | 0.01 | 0.06 | **0.13 (-)** | **0.15** |
| | | *0.63* | | *0.57* | | *0.00* | | *0.40 (-)* | | *0.10* | |
| Mixed sub. – subm. plants | D | 0.02 | 0.00 | **0.22** | 0.01 | **0.04** | 0.03 | 0.08 | 0.00 | 0.06 | 0.00 |
| | | *0.16 (-)* | | ***0.49*** | | *0.31* | | *0.37 (-)* | | *0.16* | |
| | E | **0.26** | 0.06 | 0.01 | **0.17 (-)** | 0.02 | 0.08 | 0.01 | 0.03 | **0.16 (-)** | 0.07 |
| | | *0.12* | | *0.34* | | *0.29* | | *0.19* | | *0.22 (-)* | |
| Muddy sed. - Reeds | F | 0.03 | 0.02 | **0.37** | 0.04 | 0.00 | 0.03 | 0.04 | 0.04 | 0.05 | **0.21 (-)** |
| | G | 0.01 | **0.14** | **0.13** | 0.01 | 0.05 | **0.13** | 0.00 | 0.01 | 0.02 | 0.05 |

**Legend:** ● CH$_4$   ● CO$_2$

**a** — Laboratory injections

**b** — Diffusive flux curves

**c** — Diffusive and ebullitive flux curves
Diffusive flux | Diffusive + ebullitive flux

**d** — Diffusive and ebullitive flux curves
Diffusive flux | Diffusive + ebullitive flux

**e** — Diffusive and ebullitive flux curves
Diffusive flux | Diffusive + ebullitive flux

**f** — Omitted flux curves

**Figure S1.** Raw data from the GGA showing a) a laboratory experiment where standard gas was injected in a close loop, b) diffusive flux curves in the field, c-e) diffusive and ebullitive flux curves in the field, and f) flux curves that were omitted since instrument errors or other interferences could not be ruled out.

[Figure]

**Figure S2.** CH$_4$ released per ebullition event as a function of time of day. The data is from all periods and all locations.

---

## Author Response (AR1)

**Review and community comments to manuscript "Methane ebullition as the dominant pathway for carbon sea-air exchange in coastal, shallow water habitats of the Baltic Sea" with author replies marked "AC":**

**Review #1:**

Bisander et al., report on direct observations of water-air CO2 and CH4 fluxes in the coastal Baltic Sea of Sweden. They use a floating chamber approach, with sampling campaigns covering all ice-free seasons, and use the data to produce an annual and regional synthesis. While the methods and findings do not necessarily have high novelty in of themselves, the resulting dataset is valuable in its annual coverage and application of the HELCOM ecosystem classification system. This effort improves the study's comparability with future and prior work in similar systems, and more broadly can help assess the role of these coastal habitats in nature-based climate change mitigation. The manuscript is generally quite well written, with appropriate graphics and figures. In principle, I think the study is likely a good fit for publication at Biogeosciences but have a few (mostly methodological) questions prior to committing to that decision.

AC - First of all, we would like to thank Referee #1 for their thoughtful review. Following we will address their questions and comments, with our answers marked in green:

First, a key aspect of the study is the ability to separate the ebullition fluxes from diffusive exchange. While I understand the logic for the approach that was taken, i.e. attributing the step-wise increases to ebullition, I think it could run the risk of overestimating CH4 bubble release by falsely attributing spikes in CH4 (due to sensor error, chamber effects, etc) to ebullition. No published method was cited for this approach, and no further explanation was given, say, for example based on lab or mesocosm studies, to give confidence that this method works as expected. Furthermore, the figure given in the SI is only one example flux curve, with no explanation as to whether this was a typical curve, or an especially clean one, chosen to clearly demonstrate the CH4 step-change. To address this comment, I would find it appropriate to either 1) provide some references for the method, 2) show data from when the chambers were (presumably) first tested for use in this manner, or 3) some other evidence the authors find appropriate.

AC - Thank you for pointing out the missing reference — this has now been included in the manuscript. We used a similar approach to detect ebullition in our measurements as Żygadłowska et al. (2024), and the reference is added to L 170.

We also understand the concern that spikes in $CH_4$ concentrations could be misinterpreted as ebullition, when they might instead result from analyser errors or disturbances caused by the chamber itself. The plots included in Fig. S1 in the first version of the manuscript were selected to clearly demonstrate the characteristic "step-up" pattern, but we agree they may appear too schematic, and that the manuscript could benefit from a more detailed discussion of this topic.

In response, we have expanded Fig. S1 to include additional plots — primarily of events interpreted as ebullition, but also a representative example of a diffusive measurement series for comparison. We have also added a plot showing the analyser's response to injections of standard $CH_4$ gas in a closed-loop lab test, as well as an example of field data that were excluded due to potential leakage or analyser issues. Further explanation of how peaks were evaluated and classified as ebullition has been added to the Supplementary Information Text S3. Please see the updated Supplementary Material.

Another small methodological point regards the temporal scaling. First, the daytime measured ebullition fluxes were simply scaled to the full day, despite the fact that many processes driving ebullition (atmospheric pressure deviations, boat traffic, etc) may not be evenly distributed across the "average" day. While I understand there is no way to control this, I was wondering if there might perhaps be a way to incorporate this uncertainty into the subsequent bootstrapping. If not, it might justify a brief few sentences explaining the assumptions made for 24-hour scaling, and if/when these assumptions might be broken.

AC - We have not seen a clear correlation between ebullition and time of day in our measurements, and the variability that is visible is not associated with diurnal pressure variations, where drops in pressure occurs around 4am and 4pm (SMHI, 2024). We have added a plot in Supplementary material (Fig. S3) showing the distribution of ebullition over the day, see updated supplementary material attached. In supplementary there is also a table showing the relationships between the fluxes and temperature and wind speed, now this table is updated and also includes ebullition fluxes. The relationships for ebullitive flux are almost never significant for the different locations, suggesting that it would be difficult to assess the diurnal variability in ebullition based on temperature and wind speed.

There seems to be no consistent diurnal pattern for ebullition in the literature, and while diurnal variability have been observed in rice paddies etc. this variability seems to be depending on growth stage of the plant and differ between habitats (Kajiura & Tokida, 2024; Zhu et al., 2025). Increased ebullition associated with boat traffic (Nylund et al., 2025) has been shown, however, in the sampling locations in our study the boat traffic is limited, if occurring at all.

So, while we agree that a 24h scaling most likely comes with uncertainties, these will be difficult to constrain based on these grounds, but we have added some explanation on L 189 in the end of the ebullitive flux section (2.4):

"Although diurnal variability in ebullition flux has been reported in aquatic systems, the patterns are inconsistent across habitats and plant growth stages (Kajiura & Tokida, 2024; Zhu et al., 2025). In this study, no clear relationship was found between ebullition flux and time of day or other diurnally varying factors such as temperature or wind speed (see Supplementary Material Table S1 and Fig. S2). Therefore, fluxes were scaled to 24 hours without applying corrections for potential diurnal variation."

Second, it appears in figure 2 that sampling was almost always conducted when temperatures were warmer, and wind speed was lower than the long(er) term average. Assuming water and air temperature were correlated, then it seems that sampling coincided with periods of relatively low gas solubility (thereby enhanced water-air fluxes). While I expect this effect to be small, it might be helpful to look into the temperature deviation between the sampling period and the long(er)-term average and estimate how large this bias might be. The temperature sensitivity coefficient of Takahashi et al., 1993 might provide a reasonable first approximation.

AC - Sampling was carried out during temperatures (measured by the local weather station, brought to every sampling location) that were on average 11.8 °C. During the exact periods and days of sampling, the SMHI record had an average temperature of 10.3 °C (13% lower). This suggests that first of all, there most likely is a difference in the temperature between the sampling locations and the locations of the SMHI weather stations. Between 2020 and 2022 and 8 am and 8 pm, SMHI showed a "long-term" average of 9.3 °C. So, sampling was carried out during periods and days that were 1 °C higher than the "long-term" 2020-2022 average. If this 1 °C would be representative for the water temperature as well, then the resulting solubility increase would increase the average gas transfer (and hence the average flux) by 3-4%. Each diffusive chamber measurement (both for $CO_2$ and $CH_4$) has on average a standard deviation of 30%, which has been incorporated in the annual upscaling. This suggests that the temperature influence on the solubility would only have a marginal effect on the uncertainty.

As for the wind speed, the deviation of the local measurements average compared to the specific daily averages by the SMHI and to the long-term 2020-2022 average by the SMHI agree well, suggesting that the deviation is mainly arising from differences in environmental conditions at the sampling locations vs. the SMHI weather station.

One small suggestion for figure 3 is to consider adding a second y-axis showing CO2_eq emissions, to make subplots a and b more comparable.

AC - This is a good idea, thank you! We have fixed this and also did it for Fig. 4 and 5.

References:

Kajiura, M., & Tokida, T. (2024). Diurnal variation in methane emission from a rice paddy due to ebullition. *Journal of Environmental Quality*, *53*(2), 265–273. https://doi.org/10.1002/jeq2.20553

Nylund, A. T., Mellqvist, J., Conde, V., Salo, K., Bensow, R., Arneborg, L., Jalkanen, J.-P., Tengberg, A., & Hassellöv, I.-M. (2025). Coastal methane emissions triggered by ship passages. *Communications Earth & Environment*, *6*(1), 380. https://doi.org/10.1038/s43247-025-02344-8

Zhu, T., Zhou, Y., Ju, W., Mao, Y., & Xie, R. (2025). Contributions of diffusion and ebullition processes to total methane fluxes from a subtropical rice paddy field in southeastern China. *Agricultural and Forest Meteorology*, *367*, 110504. https://doi.org/10.1016/j.agrformet.2025.110504

Żygadłowska, O. M., Venetz, J., Lenstra, W. K., van Helmond, N. A. G. M., Klomp, R., Röckmann, T., Veraart, A. J., Jetten, M. S. M., & Slomp, C. P. (2024). Ebullition drives high methane emissions from a eutrophic coastal basin. *Geochimica et Cosmochimica Acta*, *384*, 1–13. https://doi.org/10.1016/j.gca.2024.08.028

**Review #2:**

GENERAL COMMENTS

The authors report an interesting data-set of CH4 and CO2 flux measurements performed with floating chambers in vegetated shallow coastal environments in the Baltic Sea. This is a useful contribution to on-going efforts to better constrain GHG fluxes in coastal environments.

AC - First of all, we would like to thank referee #2 for their thorough and insightful review. Following we will address the questions and comments, where our replies are marked in green:

There are several places where not enough details were given and need to be added; this should be easy to address.

AC - We have gone through the specific comments and hope that they have been answered/solved in a sufficient way.

My main concern relates to the computations of the ebullitive fluxes (see below).

AC - We have provided a thorough answer to this concern where it appears under the specific comments, hopefully this has just been a misunderstanding/lack of clarity from our part and in the end, we reach the same equation.

I suggest that the authors look into temperature dependence of CH4 emissions that should provide an additional/useful angle to discuss the data.

AC - See answer where this suggestion is brought up under specific comments. These relationships have been tested and are shown in supplementary, but we have added text about this in the manuscript as well under the flux sections for $CO_2$, $CH_4$ and ebullition (section 2.2 – 2.4).

SPECIFIC COMMENTS

Please use uniform units for fluxes. In some places it's mg in other it's mgCO2 or mgCH4. Molar units are more simple and less prone to errors.

AC - We will change this so that it is written consistently mg $CO_2$ or mg $CH_4$.

There are a lot of published air-water diffusive CO2 and CH4 flux estimates in near-shore and deeper regions of the Baltic Sea as this is a densely sampled area of the world ocean. I suggest to include a discussion comparing

the air-sea fluxes in very shallow vegetated regions reported here with the more open-water (deeper) regions reported in literature. This should highlight that (at least for CH4), the shallow vegetated regions are much larger sources to the atmosphere than more open areas of the Baltic.

AC - Indeed, there are lots of published diffusive flux data for specifically $CH_4$, but also $CO_2$, in the Baltic Sea. As you say, $CH_4$ emissions are much larger in shallow, coastal waters than in the open, deeper parts of the Baltic. We have considered discussing this in more detail by comparing to studies in the deeper parts of the Baltic but considering that this would be a bit of a repetition of a discussion already brought up in both Lundevall-Zara et al. (2021) and Roth et al. (2022) we chose to focus the discussion more on only the shallow waters. However, the reviewer comment highlights that the manuscript may benefit from bringing this up in the introduction.

L 57: "At the same time, recent studies indicate they exhibit $CH_4$ fluxes up to the same order of magnitude as mangroves (Rosentreter et al., 2018) or two orders of magnitude higher than more offshore areas of the Baltic Sea (Lundevall-Zara et al., 2021)."

There does not seem to be a control site totally devoid of vegetation for comparison to the vegetated environments. This is a pity because the emissions measured here could be simply related to the muddy sediments and be unrelated to the presence of vegetation. This is a plausible hypothesis as muddy sediments are always characterized by high CO2/CH4 emissions. Such hypothesis could have only been tested with a control site devoid of vegetation. Should such a site be available from other publications, it might be useful to integrate that in the discussion.

AC - We assume that this comment only refers to the muddy sediment / reed habitat and not all habitats with vegetation in our study? All our habitats have some vegetation cover, although the coverage percentage varies quite a lot.

The chamber has sampled over both bare sediment spots in the reed beds and over submerged and emerged plants, and the resulting flux estimates are therefore "spatial averages" of the habitat. The fluxes are representing the habitat as a whole and are most likely influenced by both the muddy sediments and the vegetation. If it is the presence of vegetation or no vegetation that dictates the emissions cannot be determined by the current data set – and it is not intended to do so. The aim was to take a "whole habitat" approach, incorporating all parts of the sampled habitat and trying to better constrain an average flux.

Reeds are commonly growing in muddy sediments so the fluxes measured here would be representable for the habitat type in question – regardless of if they are mainly dictated by sediment type or by vegetation type.
We have not sampled a muddy habitat that completely lacks any vegetation. Roth et al. (2023) samples a "bare sediment" habitat, described as "soft sediment without major macrovegetation cover" which generally showed lower efflux of both $CO_2$ and $CH_4$ than our reed/mud habitat. This is mentioned in the discussion, but it does not support the hypothesis that the high emissions are only correlated to the mud.

I suggest that the authors test relations between CH4/CO2 fluxes and temperature by separating the different biomes. Air-water diffusive CH4 emissions relate to temperature in coastal environments (https://doi.org/10.1007/s10021-017-0171-7) and in lakes ebullitive CH4 emissions relate to temperature (https://doi.org/10.1007/s10533-024-01167-7). The inclusion of temperature dependence and if it differs among the biomes could be a nice addition.

AC - This has already been done for the diffusive fluxes and is shown in supplementary material Table S1. The overall low predictability of the relationships between temperature/wind/depth/salinity and the fluxes led us to the decision to only present it in supplementary materials. However, we have also added the relationships between these drivers and the ebullition flux. Again, the predictability is low and almost no relationship is significant, but we agree that it would be useful to demonstrate that the theoretically expected temperature dependence does not manifest itself clearly when the fluxes are measured directly. See updated supplementary material Table S1.

L27: I think that the control of emissions by the concentrations of gases was discovered before the paper of Gustafsson et al. (2015), the reference to Slater and Liss should suffice.

AC - Sentence changed to: "The sea-to-air exchange of these gases is governed by the water-air boundary layer conditions, which determine the gas transfer velocity and the gas saturation of surface waters (Liss & Slater, 1974)."

L 32: $CO_2$ dynamics in coastal environments are also governed by $CaCO_3$ precipitation and dissolution that can be an important component of C budget in vegetated environments (https://doi.org/10.1002/lno.11724).

AC - Elaborated: "Dissolved $CO_2$ levels are also regulated by alkalinity, which affects the balance of carbonate species and is influenced by riverine input, groundwater discharge, sediment–seawater interactions, and shelf–coastal exchange (Middelburg et al., 2020). In addition, $CaCO_3$ precipitation and dissolution can significantly modulate $CO_2$ dynamics in aquatic habitats (Champenois & Borges, 2021)."

L37: CH4 needs to be dissolved not in a bubble to "avoid microbial oxidation". I understand the point the authors want to make, but should be better phrased.
L40: It could be useful to mention that the ebullition is strongly function of depth, and that in deeper systems bubbles might come out of the sediment but fully dissolve before reaching the surface (https://doi.org/10.1029/2005JC003183).

AC - L 39 changed to accommodate both comments above: "Due to its rapid rise rate, a large fraction of free gas does not dissolve in the water column and hence avoids microbial oxidation before reaching the water-air boundary layer as ebullition flux (Mao et al., 2022), adding to the diffusive exchange across the sea-air boundary

(Hermans et al., 2024), specifically in shallow waters as ebullition increases with decreasing water depth (McGinnis et al., 2006)."

L47-50: this is a really important statement to frame the study, and requires to be backed up by references. As it stands (without references) it comes out of the blue and seems like wishful thinking.

AC - Rewriting of L 49 to L 59: "Habitat-based classifications are commonly used to upscale $CO_2$ and $CH_4$ fluxes in coastal environments, particularly in well-studied systems such as mangroves, seagrass meadows, and saltmarshes, where high productivity and sediment carbon storage have cantered research efforts (Rosentreter et al., 2023). These classifications are often tailored to tropical and subtropical environments but may not capture the full spectrum of coastal habitat diversity, especially in northern, temperate systems.

In regions like the Baltic Sea, habitats such as macroalgal beds, mixed vascular plant communities, and sparsely vegetated sediments have been shown to exchange significant amounts of $CO_2$ and $CH_4$ with the atmosphere (e.g. Lundevall-Zara et al., 2021; Asmala & Scheinin, 2023; Roth et al., 2023). These coastal shallow-water habitats vary between atmospheric annual sources of $CO_2$ (Honkanen et al., 2024) and annul sinks of $CO_2$ (Roth et al., 2023). At the same time, recent studies indicate they exhibit $CH_4$ fluxes up to the same order of magnitude as mangroves (Rosentreter et al., 2018) or two orders of magnitude higher than more offshore areas of the Baltic Sea (Lundevall-Zara et al., 2021). However, the lack of a standardized, widely adopted classification system for these habitats limits the comparability and upscaling of these flux estimates. This challenge is globally further compounded by the limited scarcity of high-resolution coastal habitat maps (Rosentreter et al., 2023)."

L57: the word "crucial" is very strong. It is certainly better but I do not think it is strictly necessary. The fluxes of CO2 and CH4 can be measured separately and then the data merged in a synthesis of data.

AC - Rephrasing: "Only a few habitats in the Baltic Sea have observations of both $CO_2$ and $CH_4$ fluxes (Asmala & Scheinin, 2023; Roth et al., 2023), and among them almost none has a full annual coverage of flux estimates, which is crucial for the effect of the sea-air exchange on radiative forcing in the atmosphere."

L60: There are several papers based on meta-analysis of data that show that the GHG emissions in CO2-equivalents are overwhelmingly dominated by CO2 and that CH4 plays a marginal role. Please rephrase "significantly" and provide numerical estimates based on literature.

AC - We assume that this comment suggests to that the total coastal oceans $CO_2$-eq. flux is overwhelmingly dominated by $CO_2$? We are a bit uncertain of what meta-analysis studies the reviewer refers to. Rosentreter et al. (2023) finds that, globally, $CH_4$ offset $CO_2$-uptake by estuaries and vegetated coastal habitats by between 6 and 16% (either 100 y or 20 y perspective) when looking at the median. However, analysing the interquartile range of the estimates suggests that $CH_4$ has the potential to offset $CO_4$-uptake by up to 58% on a 20-year timescale. Resplandy et al. (2024) finds that the total coastal ocean $CO_2$-uptake is offset by between 8 and 13% by $CH_4$

over a 100-year timescale. Significantly is of course up for interpretation, but we are not sure about "overwhelmingly dominated" either.

Further, significantly in this case does not refer to $CH_4$ having a significant role overall in the coastal GHG budget, but only highlights the fact that smaller mass quantities of $CH_4$ can play a significant role if looked at from a $CO_2$-eq perspective. But we have rephrased it to: "disproportionally". The numerical estimate will of course be dependent on the two mass fluxes in that specific habitat/location, so it will vary depending on habitat, but we have given a reference to a study with estimates for a few habitats, where the $CO_2$ uptake is offset by over 50% by $CH_4$.

Sentence: "$CH_4$ has a sustained-flux global warming potential that is 45 or 96 times as efficient as $CO_2$ on a 100- or 20-year timescale, respectively (Neubauer & Megonigal, 2015), and as such, even relatively small mass quantities of $CH_4$ can significantly contribute to the net radiative forcing added to the atmosphere by the habitat (Roth et al., 2023)."

L66: It is also possible to resolve ebullitive and diffusive emissions with eddy-covariance (https://doi.org/10.1007/s10546-018-0383-1).

AC - Sentence changed to: "The floating chamber technique, together with the eddy-covariance method, has the benefit that the flux is determined directly, therefore avoiding the use of a gas transfer velocity parametrisation and is capable of resolving the ebullition flux from the diffusive flux component (Iwata et al., 2018; Żygadłowska et al., 2024). However, since the floating chamber measurements have a high (analyser-dependent) sensitivity and small footprint, small-scale habitat differences can be resolved, and low fluxes can be included in habitat budgets."

L64-68: Only the advantages of floating chambers are cited. It could be also useful to mention some of the drawbacks of floating chambers, including artificial enhancement of turbulence and over-estimation of diffusive fluxes:

https://doi.org/10.5194/bg-12-7013-2015

https://doi.org/10.5194/bg-18-1223-2021

https://doi.org/10.4319/lo.2010.55.4.1723

and additional problems arise from changes in temperature and pressure inside the chamber during deployment:

https://doi.org/10.1061/(ASCE)0733-9372(1991)117:1(144)

https://doi.org/10.1021/es9800840

AC - Added in the end of the paragraph: "The floating chamber technique has been criticised for interfering with the boundary layer, inducing extra turbulence, as well as creating an environment within the chamber that is not representative for outside conditions in terms of wind, temperature and pressure (e.g. Mannich et al., 2019). However, taking caution in chamber design and use, these biases have been shown to have minor effects on the flux and for the chamber technique to produce similar flux values as other, non-interfering methods (Cole et al., 2010; Gålfalk et al., 2013; Lorke et al., 2015)."

L71-75: The objectives are presented in extremely general terms. It could be useful to state the working hypothesis that the paper aims to test.

AC - Elaborated aims and objectives: "In this study, we conducted year-round floating chamber experiments in shallow (<4 m) coastal habitats in the Stockholm and Trosa archipelagos of the northwestern Baltic Proper to quantify diffusive $CO_2$ and diffusive and ebullitive $CH_4$ fluxes. The objective was to constrain flux variability based on five habitat groups, including macroalgae-covered coarse sediments, sparsely to densely vegetated sands, submerged plant-covered mixed substrates, and reed-dominated muds. These habitats commonly occur along both the Swedish and Finnish Baltic Sea coast (Al-Hamdani & Reker, 2007). We hypothesized that $CO_2$ and $CH_4$ fluxes would vary systematically among these habitats, and that while within-habitat variability would occur, the boundaries for this variability would differ between the habitats. In this case, the habitat classification could help identify habitats that are the major contributors to the uncertainty associated with the coastal, shallow-water $CO_2$ and $CH_4$ flux. Further, the study aimed to quantify the relative contributions from $CO_2$ flux, diffusive $CH_4$ flux and $CH_4$ ebullition to the total $CO_2$-equivalent flux, identifying the dominant pathway for carbon-based greenhouse gas exchange in these habitats."

L90: name of plant in italic

Table 1: name of plants in italic

It could be useful to add to Table1, the average depth + the range of variation (min-max) of depth instead of only the max depth.

AC – L90 and Table 1: Fixed.

L95: it could be useful to add some extra metadata on the study sites either in Table 1 or in a separate table:

- Longitude/latitude of stations

- Exact density estimates of plants

- Total biomass estimates of plants

- Organic matter content of sediments (% org matter per weight)

- Granulometry of sediments

These infos are important to frame the study but also can be useful if the data are used by other (integration of data, comparison, etc).

AC - Longitude/latitude of sampling locations are available in the data set for anyone who wants to do spatial analysis or further research on the locations, we thought it to be redundant information to add it in the paper as well since the sites are plotted on a map.

Exact density estimates of submerged and/or emerged rooted plants and/or perennial algae has been added in Table 1.

Total biomass estimates have not been done, nor the organic content of the sediments, as neither is required for level 5 classifications in HELCOM hub.

Granulometry of the sediments are stated in the description of the classifications in Supplementary Text S1.

It could be useful to add in the supplementals photographs of the sites (plants) so readers can have a feeling of the sampled environments.

Also it could be useful to show photographs of the floating chambers.

AC - Agree, has been added to supplementary. We have not taken pictures of the exact plants at each sampling location, but we have added areal pictures of the locations.

Please specify if the floating chamber was free floating (allowed to drift) or if it was anchored or attached somehow.

AC - Current L117: "The chamber design and anchoring function is similar to those used in Schilder et al. (2016) but without an ebullition shield."

Elaborated sentence on L 120 for clarity: "Foam was attached to the downward-facing sides of the chamber for flotation, the walls submerged 1.5-2 cm into the water depending on sea state and the chamber was anchored to the seafloor."

Please specify if and how a pressure compensation was made inside the chamber on deployment. When the chamber is installed on the surface of the water it creates overpressure inside that will affect the measurements. If the flux is outgoing from the water, the over-pressure inhibits partly the outflux. Ideally after deploying the chamber, the extra pressure is vented out with a valve, and the measurements start after stabilization of pressure and closing the valve. This information is critical to determine the quality of the measurements.

AC - No pressure compensation has been done. However, this might be more of an issue for soil-chambers that are pressed down into the soil (as in Fiedler et al., 2022) than for the light-weight floating chamber used in this and similar studies. In Martinsen et al. (2018) they find a maximum pressure gradient of 0.0015 atm between ambient and headspace air of a similar chamber as ours (however, note that this maximum was found when they used an air pump to try and ventilate the chamber, not when the chamber was left as is to measure the flux, then the gradient was lower). The pressure gradient was therefore much less than 1%, which would have a neglectable impact on the gas concentration gradient between air and water and therefore also the flux.

L117: was the depth uniform for each "sampled area" ? please specify and if not provide the range of depths at each site.

AC – This information is added to Table 1.

L130: please specify at which distance was the WTW probe placed from the chambers.

AC - Changed to: "Water salinity and temperature were measured close to the surface (at ~5 cm depth, and within 10 m of the chamber deployments) with a handheld sensor 130 (conductivity WTW 340i).

L135: The LGR measures partial pressures of CO2 and CH4 not the concentrations that are calculated from Henry's law. Please clarify and rephrase.

AC - Actually, the LGR used in the study measures mole fractions of $CO_2$ and $CH_4$ that is later converted to partial pressure and then concentrations using both Dalton's law and Henry's law. Changed to: "…dC is the gas concentration change (mg $L^{-1}$), derived from the change in mole fraction as measured by the GGA…".

L 158 and Equation (4): The assumption that the ebullitive flux measurement during the 20 minute deployment of the chamber is representative of 24h does not make sense. if the chamber deployment was instead 40min, 240min or 480min it would be unlikely that you measure exactly the same ebulltive flux.

Consequently, the integration of flux at 24h needs to account for the time of the chamber deployment.

So equation (4) should be written to compute the fluxes at daily scale (24h) of each individual chamber:

metot/20 * Ch/A * 60 * 24

metot/20 provides the flux per min

* 60 converts the flux per hour

* 24 converts the flux per day

In such way the flux is integrated per day for each chamber measurement.

Then you can average all of the chamber fluxes to get a final "average" flux per day for that sampling period (that can be further integrated at seasonal scale after).

AC - Indeed, the ebullitive flux will most likely not be the same for a chamber deployment of 20, 40, 240, or 480 min. However, Eq. 4 does not scale an ebullitive flux calculated from a 20 min deployment. We think there might be a slight misunderstanding of the equation, we will try and sort this out and add comments where we can make this clearer:

Current L 185: "$me_{tot}$ is the total released $CH_4$ by ebullition during sampling in the specific period (mg)", which is derived from adding all the mg $CH_4$ released during all the single ebullition events "me" measured during a whole sampling period (month/s).

This amount is later corrected for the chamber deployment time "t", current L 168: "t is the total time that sampling was performed in a location for that month (h)." For example, if the chamber was deployed 10 times one period (month/s), the total chamber deployment time would be: (20min * 10)/60min ~ 3.33 h. This is then scaled to 24 h: $me_{tot}$*24/t (where t is 3.33h). Here we've written "month" instead of "period" in L 168 which is not consistent, we will change this to period.

Ch/A is then used for the spatial extrapolation. In the case of diffusive flux, dividing the flux in mg d-1 by A (the footprint of the chamber) would suffice since all chambers measure diffusive flux and you can assume that the whole area of the sampling location has a diffusive flux (even outside the chamber). However, for the ebullitive flux we multiply the flux in mg $d^{-1}$ by the fraction of chambers containing ebullition and then divide it by the footprint of the chamber to get a more accurate representation of the area that emits ebullition.

The suggestion by the reviewer of "($me_{tot}$/20) * (Ch/A) * 60 * 24" would overestimate the flux since it assumes that all $CH_4$ released by ebullition and detected with the chambers for the whole sampling period (month/s) was released during 20 min, when in fact it is a summation of all $CH_4$ released by ebullition during a longer period (but in different chamber deployments).

However, as I interpret the comment, I think the intent was to calculate an ebullition flux for each chamber containing ebullition, extrapolate that over 24h and then take the average ebullition flux, as we do with diffusive flux?

This would be something like $= (me_1/20A) * (1/A) * 60 * 24$, where "$me_1$" is the amount $CH_4$ in mg released by ebullition during one deployment. "Ch", the fraction of chamber deployments containing ebullition, is here removed since it is a calculation of the flux for only one chamber deployment.

To average this, you can simply take the average of all your chamber measurements that period as the reviewer suggest and where no ebullition was detected you assume zero ebullition flux. For example, maybe two chamber deployments out of 10 measured ebullition, the average would then be:

$= (((me_1 + me_2 + 8x0)/10)/20) * (1/A) * 60 * 24$

where $me_1$ and $me_2$ is the mg $CH_4$ released during the two different chamber deployments measuring ebullition (8x0 is just to represent the other 8 chamber deployments with zero ebullition). You then divide this by 10 to get the average.

This equation can be solved like:
→ $((me_1 + me_2)/200A) * 60 *24$
→ $((me_1 + me_2)/A) * 60/200 * 24$
→ $((me_1 + me_2)/A) * 24/3.33$
where $me_1 + me_2 =$ "$me_{tot}$" and $24/3.33 =$ "$24/t$"
So, it would be $= me_{tot} * 1/A * 24/t$. Which is not far from our equation in the paper.

This is how you would do it for the diffusive flux since you can assume it to be somewhat continuous. However, ebullition is likely not continuous spatially nor temporally, and with the above calculations we have technically only done a temporal averaging. To account for the likely spatial discontinuity, we multiply with the fraction of chambers containing ebullition:

$Me_{tot} * Ch/A * 24/t$, as stated in the paper.

We hope this made it clearer!

Maybe a large part of the confusion arises from L 178 – L 183, so we have changed the paragraph to: "The daily $CH_4$ emission from ebullition was extrapolated from the total mass of $CH_4$ released by ebullition as measured by all chamber deployments during the sampling period. Spatial extrapolation was conducted by assuming that the number of chamber measurements detecting ebullition for that sampling location and period relative to the total number of chamber measurements for that sampling location and period was proportional to the areal fraction of

the bay that released ebullition. The flux per m$^2$ for the period of sampling was then extrapolated over 24 hours to obtain a daily ebullition flux. The extrapolation is summarized in Eq. (4)."

L 182 : Please specify the software/packages used for the calculations

AC - Added to the beginning of the section: All statistical analyses were done using MATLAB R2022b.

L184-187: this could be transferred to the M&M

We believe it is more easily accessible in the beginning of the result section, as done in Prytherch et al. (2024) instead of dividing it up into the different subheadings of the M&M, maybe it is a question of style, we can change this if necessary.

Figure 2: can you also plot the water temperature?

AC - Only wind and air temperatures are currently plotted since they can be compared with the long-term record from the Swedish Meteorological and Hydrological Institute. But we have added a third subplot containing salinity and water temperature.

Figure 3: it is common to add the mean as a cross in box plots (in addition to median).

AC – Fixed.

L227: this is g of CO2-equi

AC - No, the CH$_4$ is g of CH$_4$, the rest are g CO$_2$-eq, have made this clearer and written it under each heading.

L 267: How was this tested? ANOVA?

AC - No, with a Kruskal-Wallis test since the distribution is not parametric, stated under 2.4 statistics L 206.

Figure 4: it could be useful to add the number of samples under each box (n=?) so reader gets an idea of representativeness.

AC - The number of samples are currently included as circle markers in the boxplots. But it is perhaps not clear what they are, will change the figure caption to: "Figure 4. Boxplots with CH$_4$ released per ebullition event (a) and CH$_4$ ebullition flux calculated for a specific location and period (b). Boxes represents 25th and 75th percentiles, line within the box is the median, whiskers are 100th and 0th percentiles. Circle markers within boxplots are single events or single fluxes. CH$_4$ released per ebullition event and CH$_4$ ebullition flux is also plotted for the specific months where it was detected (c). Months that have not been sampled are in grey."

Table 3: It could be useful to add the central value (mean or/and median) in addition to ranges. To compare different sites, what's important is the net annual flux and not simply the range. Also if the paper is justified by the idea of ultimately upscaling fluxes, then upscaling is based on a (mean or/and median) not a range.

AC - Good point, and we agree, but the majority of the cited studies lack an annual average, mainly due to only a period, or only a few months of the year being sampled which hinders upscaling to annual averages – which is as you say a motivation for this particular study. However, we've updated the table to include annual averages where available.

L 308: low organic matter certainly is not favorable for benthic respiration, but coarse sediments are very well oxygenated which is favorable for benthic respiration. The good oxygenation will to some extent compensate the low organic matter so that in the end the benthic respiration might be elevated.

AC -Indeed, but the efflux was limited in comparison to other locations, suggesting low respiration, so to include discussions on why it could hypothetically be elevated does not make sense.

L 319: Please rephrase. The habitat type is what it is and not consistent/inconsistent with literature. The measurements acquired by the authors in this habitat type were consistent/inconsistent with literature values reported by other studies at other sites in equivalent habitat type.

AC - Changed to: "While the $CO_2$ fluxes of this habitat type are less consistent with the existing literature, the clear deviation from the other habitats in this study underscores the need for a separate categorization."

L325-328: In fact, it could be even useful to use an even more simple metric such as %organic matter content in sediments (that will integrate the presence/absence of plants and any other ecosystem feature such as deposition regimes, water column productivity, depth). Available data compilations of benthic respiration (O2 consumption) or methanogenesis are based on such simple metrics.

AC - Yes, however, organic matter concentrations in the sediments are even more rare in available maps than sediment type and vegetation.

L329-332: This could be investigated in much more detail by plotting diffusive and/or ebullitive CH4 fluxes as a function of temperature. These relations can be useful to further discuss the data, as it is possible that the temperature response of CH4 fluxes is different between the habitats. At least this can be tested. Also, such relations could be useful for upscaling, and even predict how emissions could evolve in future with warming.

AC - The temperature, wind, water depth, distance to shore and salinity sensitivity of $CH_4$ and $CO_2$ diffusive fluxes are described in Table S1, due to generally low predictability of any of the relationships this has not been included in the main paper. The relationships to ebullition $CH_4$ fluxes have also been added to this table.

L334-335. This conclusion is only valid for very shallow environments (<5m) as sampled here. In deeper systems, hydrostatic pressure on sediments is less favourable for bubble formation (the CH4 remains dissolved due to higher pressure) and if bubbles are released they dissolve as they rise and never reach the surface (https://doi.org/10.1029/2005JC003183).

AC- Agree, the whole study is only relevant for very shallow waters. Will change to: "Our findings suggest that, in the coastal shallow-water zone, the variability in diffusive $CH_4$ fluxes may be of secondary importance when compared to the significant contribution of sea-air ebullition."

L342-346: This is a really important and interesting result from the study that I suggest to put forward in the abstract.

AC - Updated abstract: "Shallow coastal marine habitats are hotspots for carbon dioxide ($CO_2$) and methane ($CH_4$) exchange with the atmosphere, yet these fluxes remain poorly quantified, limiting their integration into global and regional carbon budgets. Using floating chambers, this study quantified seasonal and annual $CO_2$ and $CH_4$ fluxes in common Baltic Sea habitats, including macroalgae-covered coarse sediments, sparsely to densely vegetated sands, submerged plant-covered mixed substrates, and reed-dominated muds. Monthly average $CO_2$ fluxes ranged from $-937 \pm 161$ to $3\,512 \pm 704$ mg m$^{-2}$ d$^{-1}$, with macroalgae and reed habitats exhibiting distinct flux ranges. Apart from macroalgae, all habitats exhibited a net annual $CO_2$ efflux. Diffusive $CH_4$ fluxes varied seasonally, from $0.1 \pm 0.01$ to $26 \pm 1.5$ mg m$^{-2}$ d$^{-1}$, with peak emissions in summer. Ebullition occurred from March to October, reaching up to 232 mg m$^{-2}$ d$^{-1}$ and contributed substantially to annual carbon-based greenhouse gas fluxes in the sand, mixed-substrate, and reed habitats. Contrary to previous findings that ebullition is confined to muddy, organic-rich sediments, this study found the highest $CH_4$ ebullition in vegetated sand habitats, indicating a broader spatial extent of intense $CH_4$ release than previously assumed. Upscaling to the shallow-water (< 6 m) zone of the Stockholm archipelago yielded total $CO_2$-equivalent fluxes of between $-0.01$ and 0.2 Tg $CO_2$-eq yr$^{-1}$ (100-year timescale). For comparison, Stockholm's energy- and transport sectors emit~1.2 Tg $CO_2$-eq yr$^{-1}$, suggesting the shallow coastal zone could be a small, but non-negligible regional source for carbon-based greenhouse gases."

L 349 I do not agree with this conclusion because it only applies to very shallow parts of the ocean (<5m) that only correspond to a very small fraction of the surface area of the ocean, about 3% of the coastal ocean including the slope (down to 1000m) (https://doi.org/10.5194/hess-17-2029-2013).

AC - Sentences changed to: "The ebullition data from this study not only corroborates previous findings that CH4 ebullition dominates coastal CH4 emissions (Weber et al., 2019), but further demonstrates that ebullition could potentially be the dominant component of the coastal shallow-water carbon-based greenhouse gas flux."

L370: I agree for CH4 but not for CO2. In winter respiration might decrease but the primary production in summer will counter-act the increase of respiration due to warming. So winter CO2 emissions are usually higher

than summer even in vegetated environments (https://doi.org/10.1002/lno.11724). So the impact of temperature on respiration is only part of the story you also need to account for primary production when looking into CO2 (not the case of CH4 that indeed increases with temperature, https://doi.org/10.1007/s10021-017-0171-7).

AC - Agree to a part, $CO_2$ emission in the Baltic Sea is usually highest in October (low primary production, still warm enough for high respiration), then decreases in the winter months of December/January/February/March. Will elaborate to make this statement clearer:

"In January, the emission fluxes of both $CO_2$ and $CH_4$ at ice-free locations were high despite the coldest water temperatures of the sampling campaign, when both methanogenesis and respiration are expected to be lowest (Thamdrup et al., 1998; Yvon-Durocher et al., 2014). While wintertime efflux of $CO_2$ is expected to be higher compared to summertime when photosynthesis counter-act the respiration, peak $CO_2$ emissions are usually observed in the autumn when temperatures are still sufficient for elevated respiration, but photosynthesis has decreased (Honkanen et al., 2024; Lainela et al., 2024). However, the $CO_2$ fluxes in January were substantially higher than in October in this study."

L428: regarding "climate mitigation" you need to account for the fact that emissions from coastal (vegetated) environments remained stable since the industrial revolution so are not adding extra (or removing extra) CO2 and CH4 from the atmosphere that is increasing solely due to human perturbations of the CO2 and CH4 (fossil fuels for CO2, agriculture for CH4). What could be relevant is if the emissions from coastal environments change as a consequence of warming or eutophication or reduction of surface area (habitat destruction/restauration).

There is also the problem of the numerical importance of these fluxes. The effectiveness of "blue carbon" in mitigating global warming has been quantitatively evaluated as relatively low :

https://doi.org/10.3389/fmars.2018.00337

https://doi.org/10.3389/fclim.2020.575716

https://doi.org/10.3389/fclim.2024.1506181

doi: 10.3389/fclim.2022.853666

AC - We agree with this totally, but are not sure if it is something in the paragraph that argue against this comment? Our aim with the paragraph is simply to highlight that at the current stage of knowledge we do not know enough about the GHG-exchange to even say if the coast is a sink or a source. So, to include it as a means for "climate mitigation" would be very, very uncertain.

L428: the "sequestration" of CO2 by vegetated habitats depends on the productivity of the plants but also the

speed at which water travels on the habitat, eg the water residence time (http://dx.doi.org/10.5194/bg-2-43-2005).

AC - Sure, but again we are a bit uncertain if the reviewer thinks something should be changed as this is only a comment on the paragraph title?

L 442: the terrestrial environment is a sink of anthropogenic CO2 because we are regrowing forests in areas that were cleared of forests in the past. But fully grown "natural" (pristine) forests are not sinks of anthropogenic CO2 (https://doi.org/10.1038/NCLIMATE1804).

AC - We do not believe that the sentence at L 442 (in updated manuscript L 478) state anything regarding the sink/source status of the terrestrial environment, simply that coastal habitats are a bit more complicated than the terrestrial environment since we have the water column in between the habitat and the atmosphere.

L443-445: Only when it's warm.
Changed to "…dominant, role in the annual exchange and should be assessed…". While the ebullition only occurs and the diffusive flux is largest during the warmer months, the role it plays in the annual exchange is still substantial.

References:

Al-Hamdani, Z., & Reker, J. (2007). *Towards marine landscapes in the Baltic Sea. BALANCE interim report #10. Available at http://balance-eu.org/* . https://doi.org/10.13140/RG.2.1.3197.2726

Asmala, E., & Scheinin, M. (2023). Persistent hot spots of CO2 and CH4 in coastal nearshore environments. *Limnology and Oceanography Letters*, *published by Wiley Periodicals LLC on behalf of Assosiation for the Sciences of Limnology and Oceanography.*(n/a). https://doi.org/10.1002/lol2.10370

Champenois, W., & Borges, A. V. (2021). Net community metabolism of a Posidonia oceanica meadow. *Limnology and Oceanography*, *66*(6), 2126–2140. https://doi.org/10.1002/lno.11724

Cole, J. J., Bade, D. L., Bastviken, D., Pace, M. L., & Van de Bogert, M. (2010). Multiple approaches to estimating air-water gas exchange in small lakes. *Limnology and Oceanography: Methods*, *8*(6), 285–293. https://doi.org/10.4319/lom.2010.8.285

Fiedler, J., Fuß, R., Glatzel, S., Hagemann, U., Huth, V., Jordan, S., Jurasinski, G., Kutzbach, L., Maier, M., Schaefer, K., Weber, T., & Weymann, D. (2022). *BEST PRACTICE GUIDELINE Measurement of carbon dioxide, methane and nitrous oxide fluxes between soil-vegetation-systems and the atmosphere using non-steady state chambers*. https://doi.org/10.23689/fidgeo-5422

Gålfalk, M., Bastviken, D., Fredriksson, S., & Arneborg, L. (2013). Determination of the piston velocity for water-air interfaces using flux chambers, acoustic Doppler velocimetry, and IR imaging of the water surface. *Journal of Geophysical Research: Biogeosciences*, *118*(2), 770–782. https://doi.org/10.1002/jgrg.20064

Hermans, M., Stranne, C., Broman, E., Sokolov, A., Roth, F., Nascimento, F. J. A., Mörth, C.-M., ten Hietbrink, S., Sun, X., Gustafsson, E., Gustafsson, B. G., Norkko, A., Jilbert, T., & Humborg, C. (2024). Ebullition dominates methane emissions in stratified coastal waters. *Science of The Total Environment*, *945*, 174183. https://doi.org/10.1016/j.scitotenv.2024.174183

Honkanen, M., Aurela, M., Hatakka, J., Haraguchi, L., Kielosto, S., Mäkelä, T., Seppälä, J., Siiriä, S.-M., Stenbäck, K., Tuovinen, J.-P., Ylöstalo, P., & Laakso, L. (2024). Interannual and seasonal variability of the air–

sea $CO_2$ exchange at Utö in the coastal region of the Baltic Sea. *Biogeosciences*, *21*(19), 4341–4359. https://doi.org/10.5194/bg-21-4341-2024

Iwata, H., Hirata, R., Takahashi, Y., Miyabara, Y., Itoh, M., & Iizuka, K. (2018). Partitioning Eddy-Covariance Methane Fluxes from a Shallow Lake into Diffusive and Ebullitive Fluxes. *Boundary-Layer Meteorology*, *169*(3), 413–428. https://doi.org/10.1007/s10546-018-0383-1

Lainela, S., Jacobs, E., Luik, S.-T., Rehder, G., & Lips, U. (2024). Seasonal dynamics and regional distribution patterns of $CO_2$ and $CH_4$ in the north-eastern Baltic Sea. *Biogeosciences*, *21*(20), 4495–4519. https://doi.org/10.5194/bg-21-4495-2024

Liss, P. S., & Slater, P. G. (1974). Flux of Gases across the Air-Sea Interface. *Nature*, *247*(5438), Article 5438. https://doi.org/10.1038/247181a0

Lorke, A., Bodmer, P., Noss, C., Alshboul, Z., Koschorreck, M., Somlai-Haase, C., Bastviken, D., Flury, S., McGinnis, D. F., Maeck, A., Müller, D., & Premke, K. (2015). Technical note: Drifting versus anchored flux chambers for measuring greenhouse gas emissions from running waters. *Biogeosciences*, *12*(23), 7013–7024. https://doi.org/10.5194/bg-12-7013-2015

Lundevall-Zara, M., Lundevall-Zara, E., & Brüchert, V. (2021). Sea-Air Exchange of Methane in Shallow Inshore Areas of the Baltic Sea. *Frontiers in Marine Science*, *8*. https://doi.org/10.3389/fmars.2021.657459

Mannich, M., Fernandes, C. V. S., & Bleninger, T. B. (2019). Uncertainty analysis of gas flux measurements at air–water interface using floating chambers. *Ecohydrology & Hydrobiology*, *19*(4), 475–486. https://doi.org/10.1016/j.ecohyd.2017.09.002

Mao, S.-H., Zhang, H.-H., Zhuang, G.-C., Li, X.-J., Liu, Q., Zhou, Z., Wang, W.-L., Li, C.-Y., Lu, K.-Y., Liu, X.-T., Montgomery, A., Joye, S. B., Zhang, Y.-Z., & Yang, G.-P. (2022). Aerobic oxidation of methane significantly reduces global diffusive methane emissions from shallow marine waters. *Nature Communications*, *13*(1), 7309. https://doi.org/10.1038/s41467-022-35082-y

Martinsen, K. T., Kragh, T., & Sand-Jensen, K. (2018). Technical note: A simple and cost-efficient automated floating chamber for continuous measurements of carbon dioxide gas flux on lakes. *Biogeosciences*, *15*(18), 5565–5573. https://doi.org/10.5194/bg-15-5565-2018

McGinnis, D. F., Greinert, J., Artemov, Y., Beaubien, S. E., & Wüest, A. (2006). Fate of rising methane bubbles in stratified waters: How much methane reaches the atmosphere? *Journal of Geophysical Research: Oceans*, *111*(C9). https://doi.org/10.1029/2005JC003183

Middelburg, J. J., Soetaert, K., & Hagens, M. (2020). Ocean Alkalinity, Buffering and Biogeochemical Processes. *Reviews of Geophysics*, *58*(3), e2019RG000681. https://doi.org/10.1029/2019RG000681

Neubauer, S. C., & Megonigal, J. P. (2015). Moving Beyond Global Warming Potentials to Quantify the Climatic Role of Ecosystems. *Ecosystems*, *18*(6), 1000–1013. https://doi.org/10.1007/s10021-015-9879-4

Prytherch, J., Murto, S., Brown, I., Ulfsbo, A., Thornton, B. F., Brüchert, V., Tjernström, M., Hermansson, A. L., Nylund, A. T., & Holthusen, L. A. (2024). Central Arctic Ocean surface–atmosphere exchange of $CO_2$ and $CH_4$ constrained by direct measurements. *Biogeosciences*, *21*(2), 671–688. https://doi.org/10.5194/bg-21-671-2024

Resplandy, L., Hogikyan, A., Müller, J. D., Najjar, R. G., Bange, H. W., Bianchi, D., Weber, T., Cai, W.-J., Doney, S. C., Fennel, K., Gehlen, M., Hauck, J., Lacroix, F., Landschützer, P., Le Quéré, C., Roobaert, A., Schwinger, J., Berthet, S., Bopp, L., … Regnier, P. (2024). A Synthesis of Global Coastal Ocean Greenhouse Gas Fluxes. *Global Biogeochemical Cycles*, *38*(1), e2023GB007803. https://doi.org/10.1029/2023GB007803

Rosentreter, J., Laruelle, G., Bange, H., Bianchi, T., Busecke, J., Cai, W.-J., Eyre, B., Forbrich, I., Kwon, E., Maavara, T., Moosdorf, N., Najjar, R., Sarma, V., Dam, B., & Regnier, P. (2023). Coastal vegetation and estuaries are collectively a greenhouse gas sink. *Nature Climate Change*, 1–9. https://doi.org/10.1038/s41558-023-01682-9

Roth, F., Broman, E., Sun, X., Bonaglia, S., Nascimento, F., Prytherch, J., Brüchert, V., Lundevall Zara, M., Brunberg, M., Geibel, M. C., Humborg, C., & Norkko, A. (2023). Methane emissions offset atmospheric carbon dioxide uptake in coastal macroalgae, mixed vegetation and sediment ecosystems. *Nature Communications*, *14*(1), 42. https://doi.org/10.1038/s41467-022-35673-9

Roth, F., Sun, X., Geibel, M. C., Prytherch, J., Brüchert, V., Bonaglia, S., Broman, E., Nascimento, F., Norkko, A., & Humborg, C. (2022). High spatiotemporal variability of methane concentrations challenges estimates of emissions across vegetated coastal ecosystems. *Global Change Biology*, *28*(14), 4308–4322. https://doi.org/10.1111/gcb.16177

Schilder, J., Bastviken, D., van Hardenbroek, M., & Heiri, O. (2016). Spatiotemporal patterns in methane flux and gas transfer velocity at low wind speeds: Implications for upscaling studies on small lakes. *Journal of Geophysical Research: Biogeosciences*, *121*(6), 1456–1467. https://doi.org/10.1002/2016JG003346

Thamdrup, B., Hansen, J. W., & Jørgensen, B. B. (1998). Temperature dependence of aerobic respiration in a coastal sediment. *FEMS Microbiology Ecology*, *25*(2), 189–200. https://doi.org/10.1111/j.1574-6941.1998.tb00472.x

Yvon-Durocher, G., Allen, A. P., Bastviken, D., Conrad, R., Gudasz, C., St-Pierre, A., Thanh-Duc, N., & Del Giorgio, P. A. (2014). Methane fluxes show consistent temperature dependence across microbial to ecosystem scales. *Nature*, *507*(7493), 488–491. https://doi.org/10.1038/nature13164

Żygadłowska, O. M., Venetz, J., Lenstra, W. K., van Helmond, N. A. G. M., Klomp, R., Röckmann, T., Veraart, A. J., Jetten, M. S. M., & Slomp, C. P. (2024). Ebullition drives high methane emissions from a eutrophic coastal basin. *Geochimica et Cosmochimica Acta*, *384*, 1–13. https://doi.org/10.1016/j.gca.2024.08.028

**Community comment #1:**

What is the source of this drag coefficient of 0.0013 and can it be used across the board for shallow water? I assume that the wind has a different profile when it has previously blown over the rough landscape than when it has been able to develop a few hundred meters above the water.

AC - Thank you for your question!

The surface drag coefficient of 0.0013 was chosen based on the range of 0.0013 to 0.0015 suggested by Stauffer (1980). We selected the lower end of this range, considering that drag coefficients tend to decrease over shallow waters (<3 m) and in the presence of surface films (Hicks et al., 1974; Van Dorn, 1953) and for conformity with other studies of similar kind (e.g. Lundevall-Zara et al., 2021; Rosentreter et al., 2017). These references should, of course, have been included in the manuscript — thank you for bringing this to our attention.

As you rightly point out, the drag coefficient is also dependent on wind direction — with onshore winds likely associated with higher drag due to increased surface roughness from land. If we were to apply a drag coefficient twice the value we used (i.e., 0.0026), which falls within the upper range reported in a study of shallow, semi-enclosed lagoons (Paugam, 2021), the resulting $U_{10}$ values would increase by approximately 10%.

However, since our flux estimates are independent of wind speed, we considered a detailed sensitivity analysis of the drag coefficient to be beyond the scope of this study.

I hope this answered your question!

**References:**

References (except for those already in manuscript):

Stauffer, 1980. Windpower time series above a temperate lake. Limnol. Oceanogr. 25: 513–528. doi:10.4319/lo.1980.25.3.0513

Hicks, et al., 1974. Drag and bulk transfer coefficients associated with a shallow water surface. Bound.-Layer Meteorol. 6: 287–297.

Van Dorn, 1953. Wind stress on an artificial pond. J. Mar. Res. 12: 249–276.

Paugam, et al., 2021. Wind tides and surface friction coefficient in semi-enclosed shallow lagoons. Estuar. Coast. Shelf Sci. 257: 107406.

---

## Author Response (AR2)

**#2 Review comments to manuscript "Methane ebullition as the dominant pathway for carbon sea-air exchange in coastal, shallow water habitats of the Baltic Sea" with author replies marked "AC":**

1) The authors tested the relation to temperature of CH4 ebulltive fluxes and found no significant relationship, yet, they reported higher fluxes in summer, presumably as a response to warmer conditions. There seems to be a problem in articulating/reconciling these two findings that should be addressed.

AC – L 292 changed to: "Ebullition was confined to depths of 3 m or less, with 90% of events occurring at surface water temperatures above 11 °C. However, the magnitude of the ebullitive flux events did not consistently vary with either depth or temperature. Statistical analysis revealed inconsistency in the significance of the relationships between these variables and flux magnitude across the habitats (see Supplementary Material Table S1). Thus, while depth and temperature thresholds may govern the onset of ebullition, they appear less influential on the magnitude of the flux within the ranges where ebullition was occurring."

2) The authors need to state in the M&M that there was no pressure compensation in the chambers and they assume that over-pressure had no effect on the flux measurements.

AC – Added to L 130: "Flux measurements were conducted without compensating for potential pressure changes within the chamber as such variations have been shown to be minor (< 1 %) in light-weight floating chambers like the one used in this study (Martinsen et al., 2018). Pressure gradients of this magnitude between the ambient air and chamber environment would have a negligible effect on the flux."